# GFP-on mouse model for interrogation of in vivo gene editing

Carla Dib[1,2,18], Jack A. Queenan [3,4,5,18], Leah Swartzrock [1,2], Hana Willner[1,2], Morgane Denis [1,2], Nouraiz Ahmed[3,4,5], Fareha Moulana Zada [6,7,8], Beltran Borges[6,7,8], Carsten T. Charlesworth[2], Tony Lum[6,7,8], Bradley P. Yates[9], Caleb Y. Kwon[10], Augustino V. Scorzo [10], Scott C. Davis[10], Jessie R. Davis[3,4,5], Ran He[11,12], Jun Xie [11,12,13,14], Guangping Gao [11,12,13,14], Tippi C. MacKenzie[6,7,8], David R. Liu [3,4,5] ✉, Gregory A. Newby [3,4,5,9,15,16,17] ✉ & Agnieszka D. Czechowicz [1,2] ✉

Gene editing technologies have revolutionized therapies for numerous genetic diseases. However, in vivo gene editing hinges on identifying efficient delivery vehicles for editing in targeted cell types, a significant hurdle in fully realizing its therapeutic potential. A model system to rapidly evaluate systemic gene editing would advance the field. Here, we develop the GFP-on reporter mouse, which harbors a nonsense mutation in a genomic EGFP sequence correctable by adenine base editor (ABE) among other genome editors. The GFP-on system was validated using single and dual adeno-associated virus (AAV9) encoding ABE8e and sgRNA. Intravenous administration of AAV9-ABE8e-sgRNA into adult GFP-on mice results in EGFP expression consistent with the tropism of AAV9. Intrahepatic delivery of AAV9-ABE8e-sgRNA into GFP-on fetal mice restores EGFP expression in AAV9-targeted organs lasting at least six months post-treatment. The GFP-on model provides an ideal platform for high-throughput evaluation of emerging gene editing tools and delivery modalities.

Base editors (BEs) are powerful tools for precision genome editing as they can efficiently catalyze single-nucleotide changes at targeted genomic loci without requiring double-stranded DNA breaks (DSBs)[1,2], thereby minimizing undesired chromosomal abnormalities. Cytosine BEs (CBEs) and adenine BEs (ABEs) enable C•G to T•A and A•T to G•C transitions, respectively. Collectively, these technologies can potentially correct approximately 30% of known disease-associated variants[3]. BEs delivered via viral and non-viral methods[4–10] have successfully corrected

[1]Department of Pediatrics, Division of Hematology, Oncology, Stem Cell Transplantation and Regenerative Medicine, Stanford University School of Medicine, Stanford, CA, USA. [2]Institute for Stem Cell Biology and Regenerative Medicine, Stanford University, Stanford, CA, USA. [3]Merkin Institute of Transformative Technologies in Healthcare, Broad Institute of MIT and Harvard, Cambridge, MA, USA. [4]Department of Chemistry and Chemical Biology, Harvard University, Cambridge, MA, USA. [5]Howard Hughes Medical Institute, Harvard University, Cambridge, MA, USA. [6]Department of Surgery, University of California San Francisco, San Francisco, CA, USA. [7]The Eli and Edythe Broad Center for Regeneration of Medicine and Stem Cell Research, University of California San Francisco, San Francisco, CA, USA. [8]The Center for Maternal-Fetal Precision Medicine, University of California San Francisco, San Francisco, CA, USA. [9]Department of Genetic Medicine, Johns Hopkins University School of Medicine, Baltimore, MD, USA. [10]Dartmouth College, Thayer School of Engineering, Hanover, NH, USA. [11]Department of Genetics & Cellular Medicine, University of Massachusetts Chan Medical School, Worcester, MA, USA. [12]Horae Gene Therapy Center, University of Massachusetts Chan Medical School, Worcester, MA, USA. [13]Li Weibo Institute for Rare Diseases Research, University of Massachusetts Chan Medical School, Worcester, MA, USA. [14]Department of Microbiology, University of Massachusetts Chan Medical School, Worcester, MA, USA. [15]Department of Biomedical Engineering, Johns Hopkins University, Baltimore, MD, USA. [16]Institute for NanoBioTechnology, Johns Hopkins University, Baltimore, MD, USA. [17]Department of Molecular Biology and Genetics, Johns Hopkins University, Baltimore, MD, USA. [18]These authors contributed equally: Carla Dib, Jack A. Queenan. ✉e-mail: drliu@fas.harvard.edu; gnewby@jhmi.edu; aneeshka@stanford.edu

pathogenic mutations at therapeutically relevant efficiencies across various organs and in numerous animal models, including mice and non-human primates[11–15]. These successes have spurred recent clinical trials for the ex vivo and in vivo use of BEs with promising results to date[16,17].

Clinical in vivo transgene delivery has predominantly utilized adeno-associated virus (AAV), which couples the sustained episomal expression of a viral transgene with a tunable tropism of its twelve unique serotypes[18–20]. The naturally occurring serotypes have also been further evolved or engineered to alter their tropisms—leading to the discovery of over 100 unnatural AAV capsids used for targeted delivery[21–25]. To improve BE delivery efficiency, we have optimized the size of ABE and AAV components to package them within a single AAV vector[26]. We have shown that using a single virus to deliver BEs reduces the dose required for the desired level of editing, which would facilitate therapeutic applications. However, these single AAV vectors rely on non-*S. pyogenes* (Sp) Cas9 BEs with different PAM preferences.

To unlock the full therapeutic potential of genome-editing tools, a comprehensive evaluation of in vivo delivery efficiency, safety, and tissue targeting across various delivery vehicles is paramount. However, optimizing multiple editors and delivery strategies is arduous, costly, and time-intensive. The use of reporter mouse models has greatly sped up and simplified this process[27,28]. However, delivery strategies optimized to efficiently deliver Cre recombinase have not achieved similar efficiencies in delivering base editors[28], highlighting the need for mouse models that can directly report on base editing outcomes. A luciferase ABE-editable reporter mouse model has been reported[29] which offers real-time imaging of gene expression dynamics in live animals. Alternatively, a fluorescent reporter mouse model would allow direct visualization of EGFP using various imaging techniques, such as confocal and fluorescence microscopy, and flow cytometry, without the need for additional substrates. Notably, this would allow for single-cell resolution of editing to determine exact editing specificity even amongst rare cell types such as hematopoietic stem cells, which are 1 in ~20,000 bone marrow cells with very similar morphology to other neighboring cell types. Additionally, fluorescent protein activation would provide a permanent signal in live cells and can be visualized in fixed tissues by immunostaining. Most importantly, fluorescence permits multiplexing experiments, enabling simultaneous monitoring of multiple gene expression patterns or identification of distinct cell populations by microscopy or flow cytometry, crucial for assessing targeting efficiency in vivo.

To enable fluorescent visualization of gene editing in vitro and in vivo, we report the GFP-on reporter mouse model harboring a G-to-A nonsense mutation resulting in a premature termination codon within the EGFP sequence. To validate this model and determine the optimal single gRNA (sgRNA) to correct this mutation, we first demonstrate efficient ex vivo base editing of cells from GFP-on mouse by ABE8e mRNA and sgRNA electroporation. Next, we utilize dual AAV9 to deliver SpABE8e and the most efficient sgRNA, sgRNA1, in vivo in adult and fetal mice to observe EGFP genomic correction and restoration of EGFP fluorescence in AAV9-transduced organs in the treated mice. These findings validate the GFP-on mouse model system as an effective reporter system for detecting safe and efficient gene editing ex vivo and in vivo across various tissue types and injection routes. Additionally, we demonstrate this mouse model's utility across multiple Cas9 orthologues by performing in vivo adenine base editing using both dual AAV9-SpABE8e-sgRNA1 and single AAV9-*S. aureus* (Sa)ABE8e-sgRNA1 strategies. Lastly, through treatment of both fetal and adult mice, we show that the GFP-on mouse is a suitable reporter for adenine base editing approaches across different routes of administration and at various developmental stages.

## Results

### Generation of EGFP Q81X mouse model

To introduce an in-frame stop codon, potentially reversible with ABEs, we screened the EGFP sequence for specific codons which could be converted to a stop codon by CBE: CGA for arginine (R), TGG for tryptophan (W), and CAA/CAG for glutamine (Q). We chose three potential candidates near the 5′ end of the EGFP ORF and containing a target nucleotide positioned 12–16 bases upstream of an NGG PAM site (Q70X, Q81X, Q95X) (Supplementary Fig. 1). Of the three, Q81X (Fig. 1a) had the fewest potential bystander adenines within the ABE8e activity window (protospacer positions 3–10)[30]. Notably, Q81X was also ideally situated within the editing window of non-SpCas9 BE orthologues, including SaCas9-ABEs (NNGRRT PAM) and *S. auricularis* (Sauri) Cas9-ABEs (NNGG PAM)[31] (Supplementary Fig. 1). These orthologues are of particular interest, as their reduced size (compared to SpCas9) enables efficient packaging and in vivo delivery of a BE and sgRNA into a single AAV capsid[26].

To generate the *EGFP*^Q81X mouse model, oocytes from a female C57BL/6J mouse were fertilized in vitro with sperm from an *EGFP*⁺ male C57BL/6-Tg(CAG-*EGFP*) mouse[32]. Zygotes were microinjected with BE4max-SpCas9-NG mRNA and gRNA that introduced the Q81X (c.G241A) mutation (Fig. 1b). Five pups were born, and ear clippings were genotyped using high-throughput sequencing (HTS), revealing 9–70% editing in chimeric founder mice. These mice were then backcrossed with wild-type (WT) C57BL/6J mice. However, the resulting F1 pups harbored at most 66% Q81X, with the remaining alleles matching the WT sequence. This indicated that more than one copy of *EGFP* must be present in the original transgenic *EGFP* strain and that the first round of injections was insufficient to edit all copies.

We, therefore, conducted an additional round of BE and sgRNA microinjections into the zygotes obtained from breeding the 66% Q81X mice with WT C57BL/6J mice. Pups born from these injected zygotes were then screened for the presence of the transgene by PCR and visual green fluorescence under a UV flashlight. Pups lacking visible green fluorescence but containing the transgene were backcrossed with WT C57BL/6J mice. Finally, the *EGFP* loci of the pups resulting from this backcross were analyzed using HTS. We found 100% Q81X alleles and no nearby bystander edits. Interbreeding these mice established the first homozygous "GFP-on" reporter mice (Fig. 1c). Approximately equal proportions of male and female mice were generated from this model, which were used for subsequent experiments.

To determine the copy number of the *EGFP* sequence, droplet digital PCR was performed on DNA isolated from bone marrow cells of three GFP-on mice, with GAPDH as an internal control. The results indicated the presence of three copies of *EGFP* per GAPDH in the mouse genome, with an average of $2.97 \pm 0.07$ (Fig. 1d). This finding suggests that a homozygous GFP-on mouse would harbor six copies of *EGFP*, making the model exquisitely sensitive and efficient for assessing gene-editing strategies.

Visualization of organs from GFP-on mice post-fluorescence exposure confirmed that EGFP expression was uniformly turned off in all organs (Fig. 1e). This observation was further supported by flow cytometry results, which showed no EGFP-expressing cells in the peripheral blood, bone marrow, spleen, or liver of the mutated mice (Fig. 1f). Thorough flow cytometry analysis also revealed a complete absence of EGFP expression across all blood and bone marrow cell types from the GFP-on mouse (Supplementary Fig. 2).

### Ex vivo restoration of the EGFP expression

To correct the EGFP expression, we chose three gRNAs targeting the Q81X mutation at positions A7 (sgRNA1), A9 (sgRNA2), and A8 (sgRNA3) (Fig. 2a). *EGFP*^pm/pm fibroblasts were electroporated with SpABE8e and one of the three sgRNAs. Among all tested sgRNAs, sgRNA1 showed the highest editing efficiency, resulting in approximately 50% A>G conversion (Fig. 2b), and was selected for the subsequent experiments. To validate the choice of this sgRNA, we electroporated *EGFP*^pm/pm expanded c-Kit-enriched bone marrow cells with SpABE8e and sgRNA1. Flow cytometry analysis revealed that ~98% of the cells expressed GFP 48 h after electroporation (Fig. 2c).

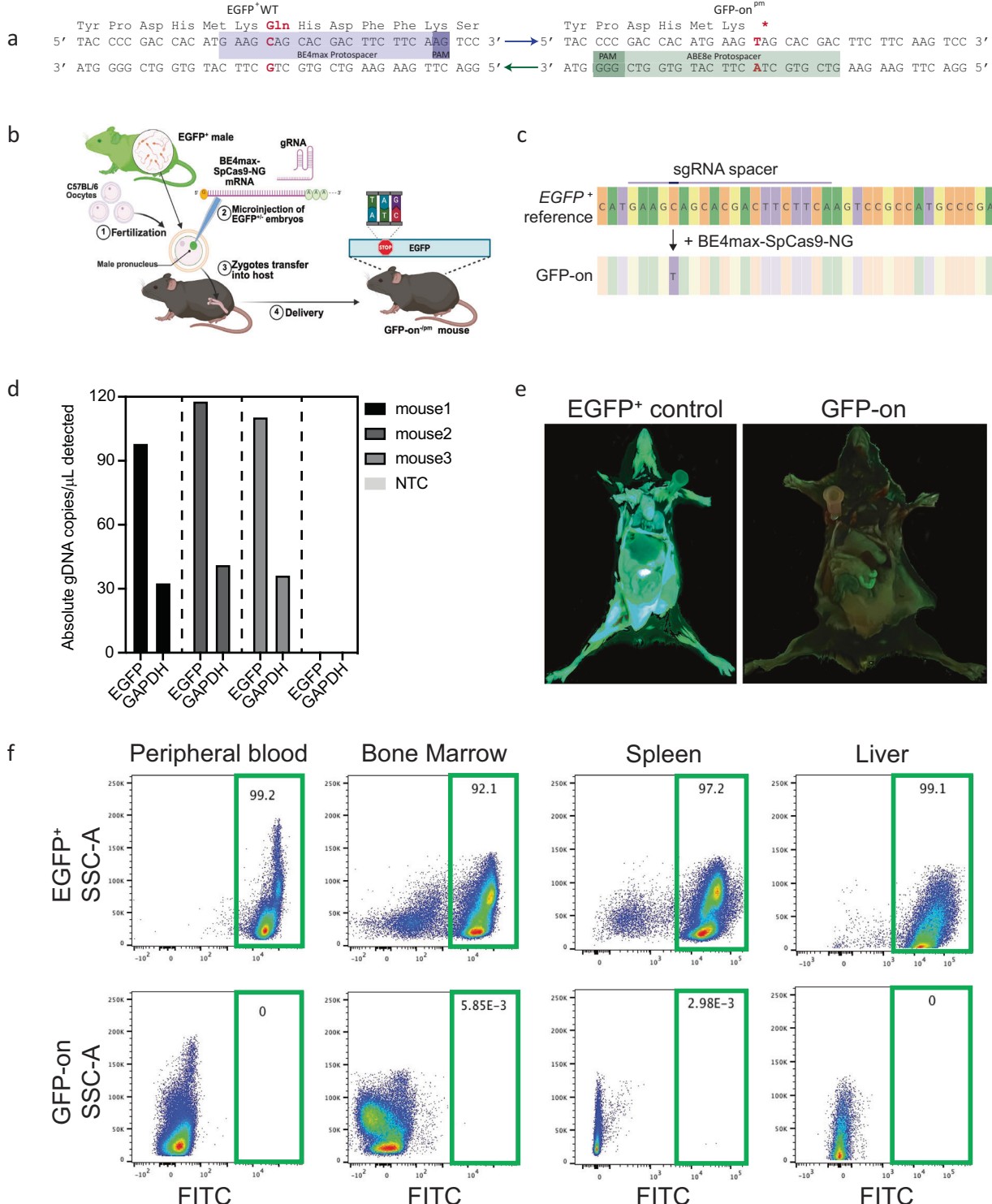

Fig. 1 | Generation of GFP-on reporter mouse model. a Sequence of the premature termination codon-containing mutant GFP-on allele. Target nucleotide in red converted by CBE to GFP-on allele and restored to WT by ABE. b Schematic detailing the generation of the GFP-on$^{-/pm}$ mouse from the EGFP$^+$ male. Created in BioRender. Czechowicz, A. (2025) https://BioRender.com/9ikb7ly. c High-throughput sequencing (HTS) of PCR-amplified genomic DNA extracted from EGFP$^{pm/pm}$ ear tissue. d Absolute number of EGFP and GAPDH copies quantified by droplet digital PCR. NTC no-template control. e Loss of EGFP fluorescence in GFP-on mice detected after opening the skin of the GFP-on mouse. Left: EGFP$^+$ mouse, right: GFP-on mouse. f Loss of GFP expression in peripheral blood, bone marrow, spleen and liver cells in GFP-on mice compared to EGFP$^+$ mice. Shown is FACS analysis of EGFP expression. Numbers in green boxes indicate % of live cell population. Source data are provided as a Source Data file.

Next, we produced dual AAV9 vectors encoding split-intein SpA-BE8e and sgRNA1 (Addgene #239016, 239017) and verified the activity of these vectors in cultured immortalized fibroblast cells from GFP-on mice. Three weeks post-transduction in fibroblasts, analysis by

microscopy imaging revealed EGFP expression from dual AAV9 delivery (Fig. 2d). Flow cytometry analysis revealed EGFP expression in approximately 2.6% of fibroblasts treated with AAV9 (Fig. 2e). HTS confirmed this statistic by detecting $3.00 \pm 0.83\%$ of edited alleles

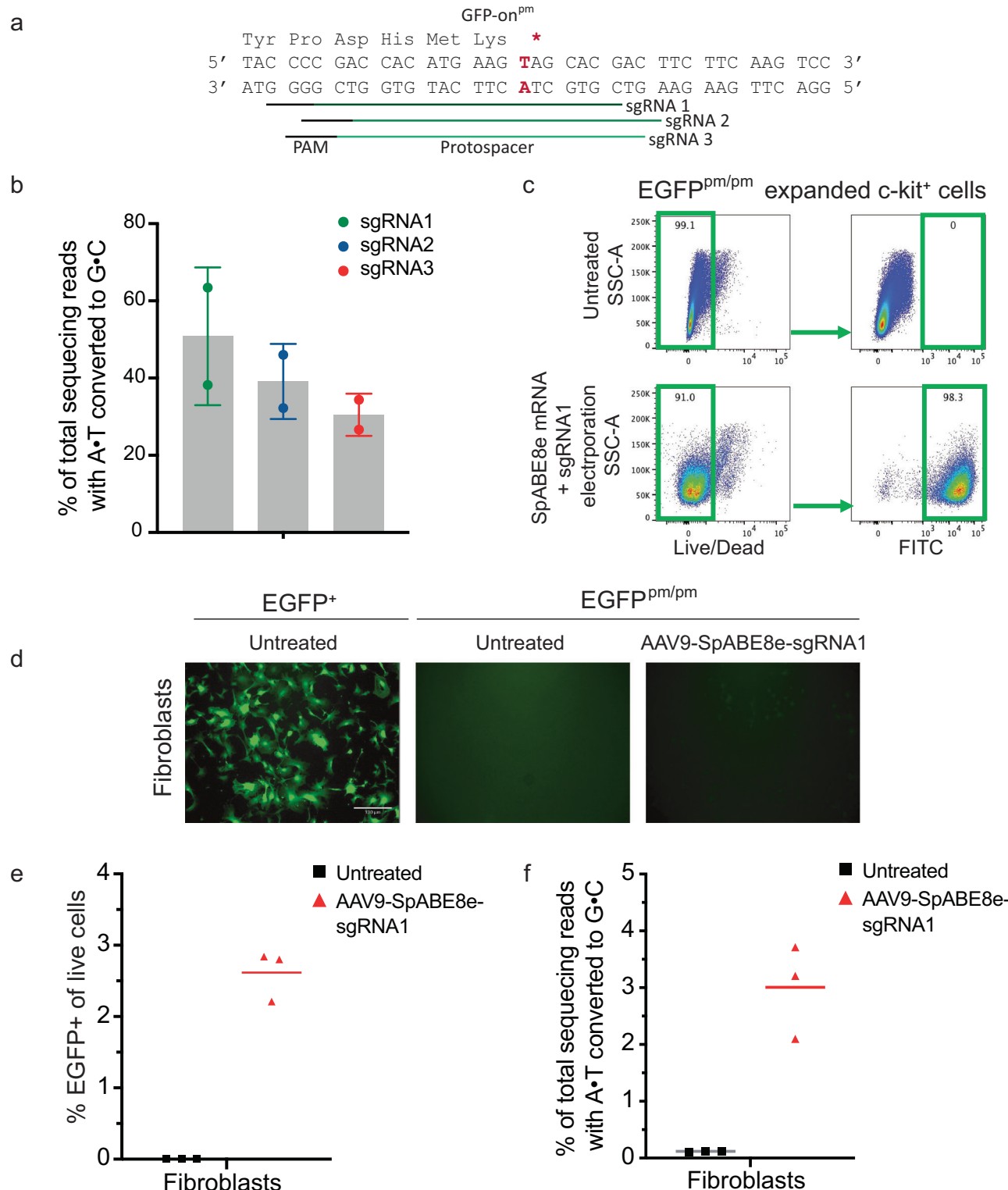

**Fig. 2 | Correction of the EGFP point mutation ex vivo. a** Schematic of EGFP locus with three guide RNAs used to correct the mutation to restore EGFP expression. **b** Highest editing efficiency observed with sgRNA1. Data indicated the percentage of EGFP correction in EGFP$^{pm/pm}$ fibroblasts electroporated with SpABE8e mRNA. $n = 2$ samples, technical replicates. Data are presented as mean with Standard Deviation (SD). **c** SpABE8e mRNA and sgRNA1 electroporation restores EGFP expression in c-Kit$^+$ bone marrow cells. Shown are FACS plots of EGFP expression in c-Kit+ bone marrow cells with or without SpABE8e mRNA electroporation. Numbers in the green plots indicate % of live cells. EGFP restoration in EGFP$^{pm/pm}$ fibroblasts transduced with dual AAV9 ($n = 3$, technical replicates) assessed by **d** fluorescence microscopy (×10 magnification, 130 μm), **e** FACS, and **f** HTS. Source data are provided as a Source Data file.

(Fig. 2f), a representative level in this AAV serotype, which is weak at transducing fibroblasts in vitro[33].

We also conducted CIRCLE-seq to nominate potential off-target editing sites that could be modified by SpABE8e and sgRNA1

(Supplementary Fig. 3). No significant off-target editing was detected at the top 29 off-target loci nominated by CIRCLE-seq in genomic DNA extracted from cells treated with AAV9 either ex vivo or in vivo (Supplementary Fig. 4).

## In vivo correction of EGFP Q81X mutation

To assess editing across multiple tissues and in hematopoietic cells, we administered dual AAV9-SpABE8e-sgRNA1 into adult GFP-on mice. A total dose of 1e12 vector genomes (vg) was delivered intravenously through combined retro-orbital (RO) (8e11) and intrafemoral (IF) (1e11 per femur) injections to maximize systemic delivery. The viral dosage comprised a 1:1 mixture (5e11 vg each) of the individual split-intein AAVs. Two weeks before the post-mortem examination, mice were fed a chlorophyll-free diet to reduce tissue autofluorescence[34]. At 3 weeks post-treatment, we assessed gene editing by EGFP expression using imaging, flow cytometry, and immunofluorescence and conducted HTS to quantify editing efficiency.

Whole-body imaging showed EGFP expression in the heart, renal tissue, brain, liver, skeletal muscles, lungs, skin, and pancreas of mice treated with SpABE8e containing AAV9 (Supplementary Fig. 5a), which was consistent with the tropism of AAV9[35]. The findings were supported by flow cytometry data showing an average of 17.5 ± 2.9% EGFP+ cells in the liver of the treated mice (Fig. 3a). Additionally, an average of 0.2 ± 0.09% and 0.04 ± 0.01% of cells expressed EGFP in the spleen (Fig. 3a) and bone marrow (Fig. 3a, Supplementary Fig. 5b), respectively, with minimal 0.02 ± 0.01% EGFP expression observed in peripheral blood cells (Supplementary Fig. 5c). Furthermore, we assessed editing in various tissues via fluorescence microscopy including brain tissue since AAV9 is known to cross the blood-brain barrier (BBB) in mice[36]. Fluorescence microscopy confirmed EGFP expression in the brain, heart, liver, and skeletal muscle (Fig. 3b). This analysis also revealed EGFP expression in the spleen, confirming the flow cytometry results. HTS analysis corroborated these findings, showing efficient editing in the heart (21.3 ± 5.4%) and moderate editing efficiency in skeletal muscle (6.77%) and spleen (3.57%) (Fig. 3c).

To assess the consistency of these results and the durable expression of EGFP, we also delivered a total of 2e11 vg or 1e11 vg of dual AAV via both RO and IF injections and evaluated EGFP expression 6 months post-injection. To reduce tissue autofluorescence, these mice were similarly fed a chlorophyll-free diet 2 weeks prior to preparing the tissues for whole-body imaging using the cryo-macrotome[34]. This approach enables detailed visualization of EGFP expression patterns in tissues and organs throughout the body, providing a quantifiable, high-resolution, three-dimensional readout of in vivo base editing efficiency throughout all tissues (Supplementary Fig. 6a)[37,38]. The results largely corroborate our findings described above, revealing EGFP expression in the brain, heart, liver, skeletal muscle, and spleen tissues of the treated mice (Supplementary Fig. 6b). As expected, the mouse injected with the higher AAV dose exhibited stronger fluorescence compared to the low AAV dose mouse (Supplementary Fig. 7).

Finally, to test the efficiency of EGFP editing using a non-SpCas9 base editor orthologue, we treated an immunodeficient GFP-on mouse with a total 1e11 vg of single AAV9 encoding SaABE8e and EGFP^Q81X sgRNA delivered via RO and IF injections. Six months post-treatment, we similarly observed EGFP expression in brain, liver, skeletal muscle, and spleen tissues (Supplementary Figs. 6, 7).

## In utero restoration of the GFP expression in GFP-on mouse fetuses

Numerous genetic hematopoietic and non-hematopoietic diseases, such as alpha-thalassemia, phenylketonuria, cystic fibrosis, Duchenne muscular dystrophy, and spinal muscular atrophy, first manifest during fetal development. Such diseases may benefit from prenatal therapy, including in utero gene editing. In this study, we aimed to further validate the GFP-on model as a platform for assessing and refining gene editing strategies for in utero gene correction. To this end, we administered 2.7e10 vg of dual AAV9-SpABE8e-sgRNA1 to nine GFP-on and eight GFP-on^−/pm fetuses via intrahepatic injection at ages E13.5-E14.5. After spontaneous delivery at term, five GFP-on pups were born. Blacklight fluorescent imaging revealed whole-body EGFP expression

in three of the five pups. EGFP expression was analyzed in these three mice at 12 weeks of age using flow cytometry. We found EGFP expression in the livers of two mice (2.5% and 18%), with no signal detected in the peripheral blood, bone marrow, or spleen (Fig. 4a). Microscopy imaging confirmed EGFP expression in the liver and brain of the same two mice, with no fluorescence observed in the spleen. Additionally, we observed EGFP restoration in the hearts and skeletal muscles of all three mice (Fig. 4b). HTS analysis similarly revealed EGFP editing in the brain, heart, liver, and skeletal muscle (Fig. 4c), with no editing in the other assessed organs.

Of the eight GFP-on^−/pm injected fetuses, six pups were born, four of which had EGFP expression (Supplementary Fig. 8a). We evaluated EGFP expression in one of these mice 6 weeks post-birth using flow cytometry and found no EGFP expression in the peripheral blood, bone marrow, or spleen. However, we found that EGFP expression was restored in the liver (Supplementary Fig. 8b). Immunofluorescence results confirmed EGFP expression in the hepatocytes stained with hepatocyte nuclear factor-4 alpha (HNF4A) antibody. Staining with the neuronal nuclei (NeuN) antibody revealed the localization of EGFP within the brain neuronal cells. Finally, staining with the laminin antibody revealed EGFP with and within cardiomyocytes and muscle fibers (Supplementary Fig. 8c). To assess the durability of EGFP expression, we examined EGFP expression levels in the remaining fluorescent animals at 6 months of age. The EGFP expression pattern observed in these mice was comparable to the mouse assessed at 6 weeks post-birth, suggesting long-lasting expression as expected given editing of the genome (Supplementary Fig. 8b–d). In all three mice, EGFP expression was detected in the brain, heart, liver, and skeletal muscle, while none was observed in the spleen (Supplementary Fig. 8d). HTS data analysis confirmed editing in the following organs: heart (35.0 ± 6.1%), muscle (7.4 ± 5.1%), and liver (4.3 ± 3.9%) (Supplementary Fig. 8e). As each cell harbors six copies of EGFP, we would anticipate that the percentage of edited alleles would generally be lower than the percentage of EGFP+ cells that can be identified through flow cytometry, unless all alleles in an EGFP+ cell are edited in which case these values should be equal.

## Discussion

By introducing a premature termination codon (PTC) into a pancellular, constitutive EGFP gene expressing mice, we generated a new fluorescent protein reporter mouse model to facilitate head-to-head comparisons of BE delivery vehicles. We selected the PTC−introduced using cytosine BE−based on its ability to be corrected by the current suite of therapeutic ABEs, including S. pyogenes (Sp), S. aureus (Sa), and S. auricularis (Sauri) ABEs.

Cre recombinase reporter strains, such as Ai14, are commonly used mouse models in the protein and nucleic acid delivery field, which contain a loxP-flanked STOP cassette downstream of a CAG promoter and upstream of a fluorescent reporter (tdTomato for Ai14) designed so Cre recombination will turn on fluorescent protein expression. The GFP-on mouse is preferential to Cre reporter strains for several reasons. First, the Cre reporter strains have been noted to exhibit variable levels of Cre-independent reporter expression. By contrast, the untreated GFP-on mouse did not exhibit leaky EGFP expression in any of the tested cell types or organs by flow cytometry or under blacklight. Second, the GFP-on mouse should directly mirror therapeutic adenine base editing efficiency that could be achieved in targeted cell types using a standard or bespoke delivery vehicle. In many cases, the delivery of Cre recombinase to mouse tissues has tended to be substantially more efficient than base editor delivery by identical methods[28,39,40], perhaps due to the smaller size of Cre recombinase permitting better expression, the intrinsic sequence recognition of Cre that requires neither co-delivery nor binding of a guide RNA, by innately higher activity of Cre editing efficiency, or a combination of these three factors. Reporting on the delivery of the base editor directly will enable more direct optimizations of therapeutic cargo.

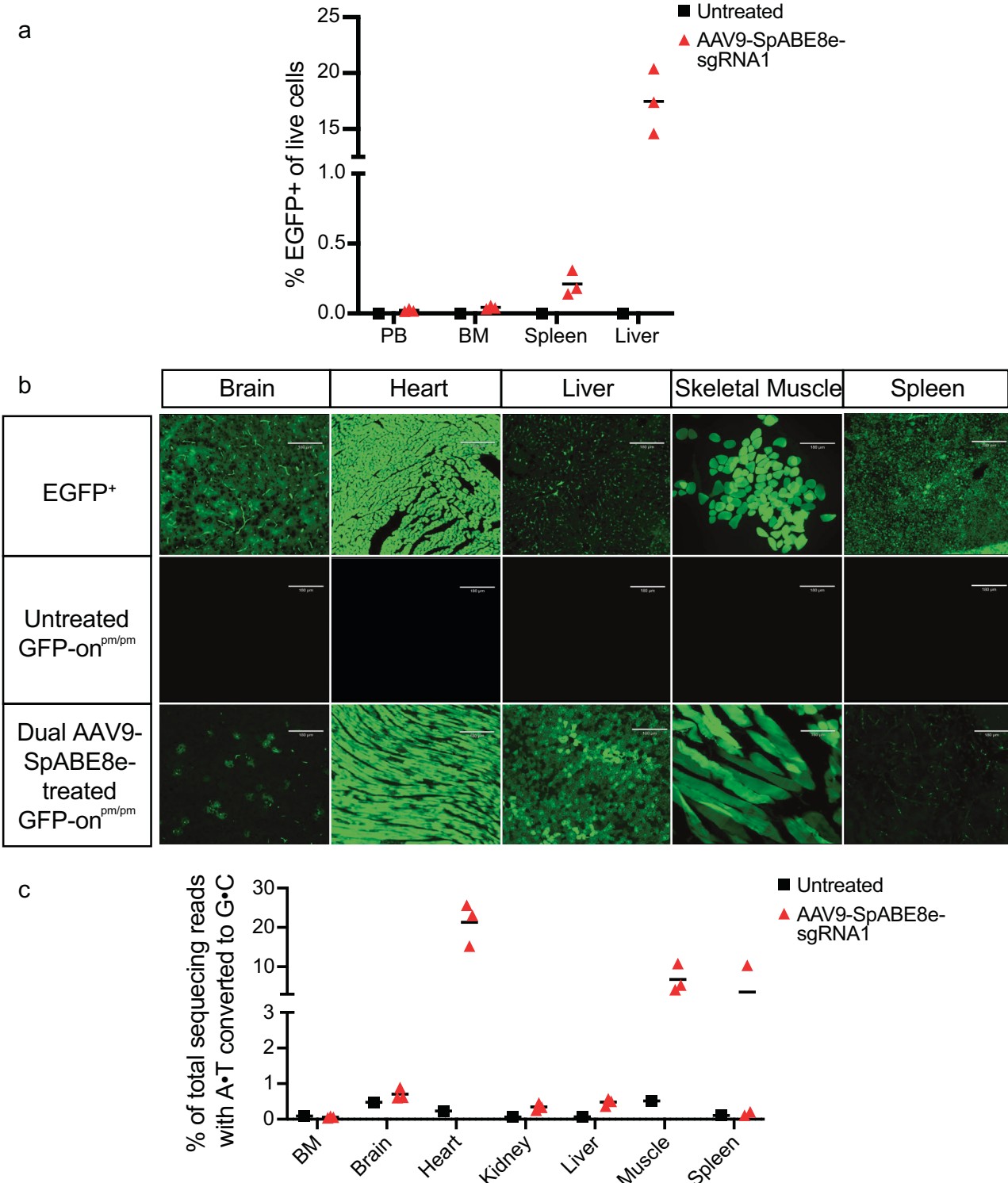

**Fig. 3 | Correction of the EGFP point mutation in adult mice.** Assessment of EGFP expression in various organs of GFP-on$^{pm/pm}$ mice ($n = 3$, biological replicates) 3 weeks post systemic in vivo treatment with dual AAV9 containing SpABE8e-sgRNA1 via **a** flow cytometry, **b** fluorescence microscopy (×10 magnification, 100 μm), and **c** HTS. Data are presented as mean. PB peripheral blood, BM bone marrow. Source data are provided as a Source Data file.

We used this model to recapitulate the known tropism of systemically injected dual AAV9 expressing SpABE8e and sgRNA. These observations were also reliably recapitulated using whole-body fluorescent cryo-imaging to visualize the distribution of successfully edited cells at the micron-scale throughout the entire mouse—a technique that eliminates selection bias for commonly screened tissues of interest, which is critical for evaluating emerging delivery vectors. We also showed that in utero treatment of fetuses via intrahepatic injection of AAV9 resulted in a biodistribution pattern similar to that observed via retro-orbital injection in adult mice, apart from a lack of splenic editing, with sustained expression lasting over 6 months in many different organs and tissue types. Importantly, this was accomplished using 37× less virus, which could make systemic treatment more feasible and less expensive. While not assessed in this study,

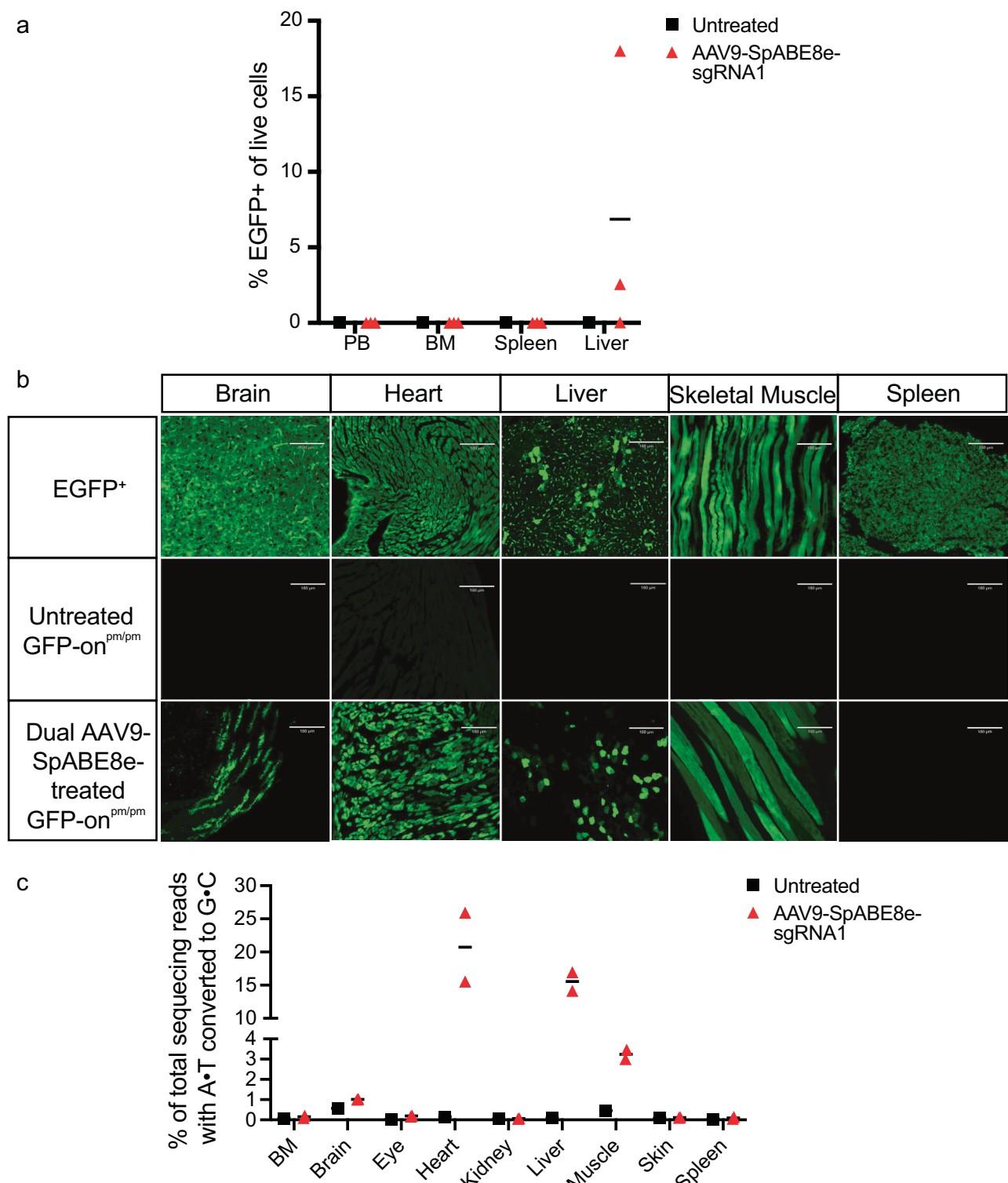

**Fig. 4 | Correction of the EGFP point mutation in fetal mice.** Assessment of EGFP expression in various organs of GFP-on^pm/pm mice ($n$ = 3, biological replicates) post-treatment in utero with dual AAV9 containing SpABE8e-sgRNA1 at **a** 8–12 weeks post-birth in various organs via **a** flow cytometry, **b** fluorescence microscopy (×10 magnification, 100 μm), and **c** HTS. Data are presented as mean. PB peripheral blood, BM bone marrow. Source data are provided as a Source Data file.

prior in utero AAV9 treatments have reported minimal leakage into maternal tissues with multiple studies in pregnant mice and lambs finding no detectable transgene expression in maternal tissues, though elevated bilirubin and anti-AAV antibody levels have occasionally been observed indicating some low level of exposure is possible[41–44].

We also demonstrated that the primary c-Kit⁺ bone marrow cells from this mouse model can be harvested and edited ex vivo with high efficiency by mRNA electroporation to restore EGFP expression. Likewise, fibroblasts could be isolated from these mice and edited ex vivo with dual AAV9. Inducible luminescence mouse models are limited by their inability to simultaneously detect editing efficiency

and cell surface markers (as luminescence signals are incompatible with flow cytometry analysis). In contrast, our GFP-on mouse model is compatible with flow cytometry analysis, easing the assessment of complex, diverse tissues such as bone marrow. Indeed, we demonstrated that individual subpopulations of hematopoietic stem and progenitor cells can be separately assessed and quantified for EGFP fluorescence, highlighting the unique capability of this mouse model to enable rapid measurement of viral transduction selectivity in mixed cell populations.

We also show that the GFP-on mouse model is a powerful tool for studying the in vivo delivery of BEs across tissues. By directly coupling base editing efficiency to fluorescent protein expression, the GFP-on mouse model enables non-destructive, flow cytometry-based quantification of base editing efficiencies in sortable cell subpopulations. Nearly any BE delivery vehicle can be assessed for tissue-level biodistribution or single-cell level tropism using this model. Unlike its predecessors, this model can be targeted by *S. aureus* and *S. auricularis* ABEs, which are small enough to be packaged into a single AAV, thereby broadening the delivery methods suitable for the GFP-on model to match the current suite of therapeutic base editors. We expect that this reporter mouse model will enable faster and more detailed analysis of the cell type-specificity of therapeutic gene editing delivery methods with an unprecedented degree of genome editor cargo modularity.

Further, by harboring 6 copies of mutated *EGFP*, the GFP-on mouse model increases the sensitivity of detecting gene editing events by providing more targets for correction. This enhances the likelihood of successful editing, even in cases of lower efficiency. Additionally, it amplifies the fluorescent signal upon correction, making it easier to detect and quantify editing outcomes. The elevated copy number also decreases the percentage of edited alleles in this model relative to the percentage of edited cells. This effect becomes most apparent in vivo due to the challenge of achieving high transgene expression in vivo relative to ex vivo cultured cells. Therefore, the percentage of EGFP⁺ cells is a more translatable indicator of base editing efficiencies across other protospacers.

In summary, the mouse model developed in this study enables efficient and robust assessment of precision gene editing technologies and in vivo delivery methods. It also facilitates testing different administration routes or combinations with other concurrent therapeutic interventions. Moreover, it enables rapid assessment across various developmental stages, determining the optimal timing for intervention. The GFP-on mouse model is available openly upon request.

## Methods

### Ethics statement

All experiments were performed according to guidelines and approvals established by the Administrative Panel on Laboratory Animal Care of Stanford University (Protocol# 32459), the Broad Institute Institutional Animal Care and Use Committee (Protocol# 0048-04-15-2) and the Institutional Biosafety Committee (Protocol# IBC2017-00143).

### Mice

All mice were housed in a room maintained on 12 h light-dark cycle. The ambient temperature was maintained at 20–24 °C with relative humidity between 40 and 60%. Mice had ad libitum access to food and water and were housed in individually ventilated cages with appropriate bedding and enrichment. Two weeks before euthanasia, mice feed was changed to a chlorophyll-free diet (MP Biomedicals, Santa Ana, California) to reduce tissue autofluorescence.

### Generation of GFP Q81X mouse model

GFP-on mice were generated by injecting zygotes from WT C57BL/6J (JAX strain #000664) oocytes fertilized in vitro with sperm from C57BL/6-Tg(CAG-EGFP)131Osb/LeySopJ mice (JAX strain #006567)

with BE4max-SpCas9-NG mRNA purchased as a custom product from Trilink Biotechnologies and gRNA ordered from Synthego to direct the editor to introduce the c.G241A point mutation. The guide RNA spacer sequence was: GAAG**C**AGCACGACUUCUUCA and contained Synthego's standard chemical modifications, 2′-*O*-methyl modifications in the first three and last three nucleotides, and phosphorothioate bonds between the first three and last two nucleotides. The zygotes were then transferred into pseudo-pregnant C57BL/6J mice. HTS confirmed the introduction of the desired point mutation in 9–70% of alleles in chimeric founders, and after backcrossing to C57BL/6J mice, at most 66% allelic editing was observed, indicating incomplete editing of all *EGFP* copies. A subsequent round of BE4max-SpCas9-NG mRNA and gRNA microinjection was conducted in zygotes for which the oocyte harbored 66% Q81X and the sperm was WT C57BL/6J, resulting in mice that bred true with C57BL/6J mice, yielding pups with 100% Q81X alleles. GFP-on⁻/ᵖᵐ animals were intercrossed to obtain *EGFP* homozygous mutants (GFP-onᵖᵐ/ᵖᵐ). qPCR was used to differentiate between heterozygote and homozygous mice.

### Droplet digital PCR determination of EGFP copy number in reporter mouse genome

Genomic DNA from 3 *EGFP* homozygous mice was isolated from bone marrow cells with the NucleoSpin®Tissue kit (Macherey-Nagel, Allentown, PA). 2 ng of genomic DNA was added per 40 μL digital PCR reaction, and negative control with no template DNA was run simultaneously with mouse samples. Digital PCR was run on a QIAcuity instrument using the QIAcuity probe PCR kit (Qiagen, Germantown, MD) according to the manufacturer's instructions. In addition to the template DNA and kit components, these reactions included a 1x digital FAM-labeled PCR primer/probe mix for mouse GAPDH as a multiplexed internal control (Bio-Rad, Hercules, CA, assay ID dMmuCNS133125454). While the exact design of the primers and probes in this proprietary assay are not public, Bio-Rad shares the target sequence of this assay as: CCAATAAAGATACATGCACAAA AGTTGATTGAGCCTGCTTCACCTCCCCATACACACCCTCCCTCCCCC AACACCGCATTAAAACCAAGGAGAGGTGGGTGCAGCGAACTTTATTG ATGGTAT. While several homologous pseudogenes similar to GAPDH exist in the mouse genome, an NCBI BLAST search of this target sequence within the mouse genome revealed no other site with more than 75% homology, and thus we expect this assay to be specific for a single target and allow us to normalize copy number calculations to a gene present in two copies in a diploid genome. Reactions also included a custom HEX-labeled PCR primer/probe mix for GFP copy number quantification: Forward primer 5′-ATCTTCTTCAAGGACGACGG CAAC-3′; Reverse primer 5′-AGCTCGATGCGGTTCACCAG-3′; probe 5′-/ 5HEX/TCGCCCTCG/ZEN/AACTTCACCTCGGCGC/3IABkFQ/-3′ (800 nM of each primer and 400 nM of the probe final concentration in each reaction). HindIII (New England BioLabs, Ipswich, MA) was added to a final concentration of 0.25U/μL and BssSI (New England BioLabs, Ipswich, MA) was added to a final concentration of 0.125U/μL to cleave genomic DNA before partitioning; the *EGFP* gene contains a BssSI site outside of the amplified target. 40 μL reactions were incubated for 20 min at 37 °C to permit digestion of genomic DNA, then added to a 24-well QIAcuity nanoplate (Qiagen, Germantown, MD). The digital PCR cycling program consisted of 95 °C for 2 min, followed by 40 cycles of 15 s of denaturation at 95 °C and 30 s of combined annealing and extension at 60 °C. The concentrations of GAPDH genomic target and *EGFP* genomic target were calculated by the QIAcuity instrument software. The genomic *EGFP* copy number was normalized to the genomic GAPDH copy number of the same reaction to calculate the relative number of *EGFP* copies in the mouse genome.

### High-throughput sequencing of genomic DNA

Genomic DNA was isolated with the NucleoSpin®Tissue kit. 5–50 ng was used as input for the first of two PCR reactions. Genomic loci were

amplified in PCR1 using NEBNext polymerase (New England Biolabs, Ipswich, MA). PCR1 primers for genomic loci are listed in Supplementary Table 1. PCR1 was performed as follows: 95 °C for 3 min; 25–35 cycles of 95 °C for 15 s, 1 °C for 20 s, and 72 °C for 30 s; 72 °C for 2 min. PCR1 products were confirmed on a 1% agarose gel. 1 μL of PCR1 was used as input for PCR2 to install Illumina barcodes. PCR2 used 9 cycles of amplification using NEBNext (New England Biolabs, Ipswich, MA). Following PCR2, samples were pooled and gel-purified in a 1% agarose gel using a Qiaquick Gel Extraction Kit (Qiagen, Germantown, MD). Library concentration was quantified using Qubit High-Sensitivity Assay Kit (Thermo Fisher, Houston, TX). Samples were sequenced on an Illumina Miseq instrument (paired-end read, read 1: 220–280 cycles, read 2: 0 cycles) using an Illumina Miseq 300 v2 Kit (Illumina, San Diego, CA).

### High-throughput sequencing data analysis

Sequencing reads were demultiplexed using the Miseq Reporter software (Illumina, San Diego, CA) and were analyzed using CRISPResso2 under batch analysis mode[45]. Reads were filtered by minimum average Phred score (Q > 30) prior to analysis. The following window parameters were used: -w 20 -wc 10. Base editing efficiencies are reported as the percentage of sequencing reads containing a given base conversion at the specified position. GraphPad Prism 10 was used to generate bar plots of sequencing data.

### Generation of stable EGFP^pm/pm and EGFP⁺ fibroblast lines

Tail tissues from GFP-on and EGFP⁺ mice were digested with 500 μg/mL of Collagenase P (Sigma-Aldrich, St. Louis, MO) for 2 h at 37° C. Single-cell suspension was obtained by grinding the digested tissues through a 70 μm strainer. The cells were washed and centrifuged at 200×$g$ for 5 min then cultured in Dulbecco's Modified Eagle Medium (DMEM) low glucose, pyruvate (1X) (Gibco, Grand Island, New York) supplemented with 10% Fetal Bovine Serum (FBS) (Sigma-Aldrich, St. Louis, MO) and 1% Penicillin/streptomycin (P/S) (Gibco, Grand Island, New York) at 37 °C with 5% CO$_2$. Hek293T producer cells were seeded at a density of $7 \times 10^6$ cells in a 10 cm plate (Corning, Glendale, Arizona) in high glucose DMEM (Gibco, Grand Island, New York), supplemented with 10% FBS, 1% P/S, and 1% L-glutamine (Gibco, Grand Island, New York). 24 h post culture, cells were transfected with the following constructs: p1321 HPV-16 E6/E7 (5.25 μg) (Addgene, Watertown, MA), VSV-G (0.75 μg) and MSCVgag/pol (1 μg), using the TransIT-239 transfection reagent (Mirus Bio, Madison, WI). The media was replaced 16 h post-transfection. The supernatant was harvested, centrifuged at 500×$g$ for 5 min, then filtered through a 0.45 μm PVDF filter. Fibroblasts were seeded at a density of $1 \times 10^6$ cells in a 10 cm plate. 24 h post culture, they were transduced with the supernatant containing retrovirus in the presence of polybrene at 8 μg/mL (Sigma-Aldrich, St. Louis, MO). 48 h post-transduction, Geneticin antibiotic (G418) (Invitrogen, Waltham, MA) was added to the media at a concentration of 2500 μg/μL for 11 days to select the immortalized cells.

### Electroporation of EGFP^pm/pm fibroblasts

EGFP^pm/pm primary fibroblasts were electroporated with a Lonza 4D-nucleofector using the electroporation program DS-150. Ten million cells per mL in 20 μL P2 buffer were electroporated with 3 μg of ABE8e-NGG mRNA and 50pmol of one of the three synthetic sgRNA (Guide1: 5′-CGUGCU**A**CUUCAUGUGGUCG-3′, Guide2: 5′-GUCGUGCU**A**CUUCAUGUGGU-3′ and Guide3: 5′-UCGUGCU**A**CUUCAUGUGGUC-3′). The guides contained 2′-$O$-methyl modifications in the first three and last three nucleotides and phosphorothioate bonds between the first three and last three nucleotides and were purchased from Synthego[46]. Three days post-electroporation, the media was removed, and cells were washed with 1×Phosphate-buffered saline (PBS) (Thermo Fisher, Houston, TX). After PBS aspiration, genomic DNA was extracted from adhered cells by the addition of 0.05% SDS, 10 mM Tris pH 8.0, 2 U/mL

proteinase K (New England BioLabs, Ipswich, MA). Cells were incubated in this buffer for 1 h at 37 °C, followed by heat-inactivation of proteinase K at 80 °C for 30 min. One microliter of this extract (about 100 ng of DNA) was used as a template for PCR to amplify the targeted GFP transgene for HTS to assess editing efficiency.

### Electroporation of EGFP^pm/pm c-Kit⁺ expanded cells

GFP-on mice were euthanized, and bone marrow cells were collected by flushing the femurs. After 20 min of staining with anti-c-Kit (CD117) antibody (Miltenyi Biotech, Auburn, AL), the cells were enriched by MACS (Miltenyi Biotech, Auburn, AL) following the manufacturer's instructions. c-Kit-enriched EGFP^pm/pm cells were cultured in HSC expansion media: F12 (Gibco, Grand Island, New York), 0.1% PVA (Sigma-Aldrich, St. Louis, MO), 1% P/S/glutamine (Gibco, Grand Island, New York), 1% ITSX (Gibco, Grand Island, New York), 100 ng/mL mTPO (Peprotech, Cranbury, NJ), 10 ng/mL SCF (Peprotech, Cranbury, NJ)[47,48]. Three weeks post-expansion, cells were electroporated with a Lonza 4D-nucleofector using the electroporation program EO-100. Twenty million cells per mL in 20 μL P3 buffer were electroporated with ABE8e-NGG mRNA and sgRNA1 (Synthego, Redwood City, CA) at 2 μg/μL and 8 μg/μL, respectively. Electroporated cells were recovered in HSC expansion media. 48 h post-electroporation, they were washed and stained with blue viability dye (Thermo Fischer Scientific, Houston, TX) following the manufacturer's instructions. GFP expression was assessed by flow cytometry using a FACSAria II cell sorter (BD Biosciences, Franklin Lakes, New Jersey) and analyzed in FlowJo v10 (Tree Star, Ashland, OR).

### AAV9 production and concentration

AAV9 was produced as previously described[4]. Guide RNA spacer sequences (SpABE8e: 5′-CGTGCT**A**CTTCATGTGGTCG-3′, or SaABE8e: 5′- AGTCGTGCT**A**CTTCATGTGGT- 3′) were cloned into AAV transfer vectors by Gibson assembly. HEK293T clone 17 cells (ATCC CRL-11268) were maintained in DMEM plus GlutaMax (Thermo Fischer Scientific, Houston, TX) supplemented with 10% (v/v) FBS without antibiotic in 150 mm dishes (Thermo Fischer Scientific, Houston, TX) at 37 °C with 5% CO$_2$ and passaged every 2–3 days. Cells used for production were seeded at a density of $9.12 \times 10^6$ cells per dish 1 day prior to polyethyleneimine transfection with 5.7 μg AAV genome plasmid, 11.4 μg pHelper (Clontech, Mountain View, CA), and 22.8 μg rep-cap plasmid per plate. 18–24 h after transfection, media was exchanged for DMEM plus GlutaMax with 5% (v/v) FBS. 4 days after transfection, cells and media were collected using a cell scraper (Corning, Glendale, Arizona), pelleted by centrifugation for 2000×$g$ for 10 min, resuspended in 500 μL hypertonic lysis buffer per plate (40 mM Tris base, 500 mM NaCl, 2 mM MgCl$_2$, 100 U/mL salt-active nuclease (ArcticZymes; Wayne, PA)) and incubated at 37 °C for 1 h to lyse the cells. The media was decanted and combined with a 5x solution of 40% poly(ethylene glycol) 8000 (PEG 8k) (Sigma-Aldrich, St. Louis, MO) in 2.5 M NaCl (final concentration 8% PEG, 500 mM NaCl). This solution was incubated on ice for 2 h and centrifuged at 3200×$g$ for 30 min. The viral pellet was resuspended in 500 μL hypertonic lysis buffer per plate and added to the cell lysate. Crude lysates were then incubated at 4 °C overnight.

Cell lysates were clarified by centrifugation at 2000×$g$ for 10 min and added to Beckman Quick-Seal tubes via 16-gauge 5" disposable needles (Air-Tite, Virginia Beach, VA). A discontinuous iodixanol gradient was formed by sequentially floating layers of 9 mL 15% iodixanol in 500 mM NaCl and 1X PBS-MK (1X PBS, 1 mM MgCl$_2$, 2.5 mM KCl), 6 mL 25% iodixanol in 1X PBS-MK, and 5 mL each of 40 and 60% iodixanol in 1X PBS-MK. Phenol red at a final concentration of 1μg/mL was added to the 15, 25, and 60% layers to facilitate identification. Ultracentrifugation was performed using a Ti 70 rotor in an Optima XPN100 ultracentrifuge (Beckman Coulter, Brea, CA) at 475,900×$g$ (67,000 r.p.m.) for 1 h and 15 min at 18 °C. Immediately following centrifugation, 3 mL of solution was withdrawn from the 40−60% iodixanol interface using an 18-gauge

needle, exchanged into cold PBS containing 0.001% F-68 via PES 100-kD MWCO columns (Thermo Fischer Scientific, Houston, TX), and concentrated. The concentrated AAV solution was sterile filtered through a 0.22 μm filter, quantified by qPCR (AAVpro titration kit version 2; Clontech, Mountain View, CA), and stored at 4 °C until use.

### Transduction of EGFP$^{pm/pm}$ fibroblasts with dual AAV9
Immortalized EGFP$^{pm/pm}$ fibroblasts were plated in fibroblast culture media in 96-well plates (Corning, Glendale, Arizona) at a density of 100 K cells per well. AAV9-SpABE8e was added to the cells at an MOI of 2e6. Three weeks post-transduction, EGFP expression was assessed by microscopy imaging using the Echo confocal microscope (BICO, San Diego, California), by flow cytometry using FACSymphony A5SE (BD Biosciences, Franklin Lakes, New Jersey), and by HTS.

### CIRCLE-seq off-target editing analysis
CIRCLE-seq was conducted to experimentally nominate candidate off-target base editing sites for the editing strategy that activates the GFP-on reporter. Mouse genomic DNA was isolated from cultured mouse N2A cells using the QIAgen Gentra PureGene kit. CIRCLE-seq was performed as previously described[39]. Briefly, purified genomic DNA was sheared with a Covaris S2 instrument to an average length of 300 bp. The fragmented DNA was end-repaired, A-tailed and ligated to an uracil-containing stem-loop adapter, using the KAPA HTP Library Preparation Kit, PCR Free (KAPA Biosystems). Adapter ligated DNA was treated with Lambda Exonuclease (NEB) and E. coli Exonuclease I (NEB) and then, after purification, with USER enzyme (NEB) and T4 polynucleotide kinase (NEB). Intramolecular circularization of the DNA was performed with T4 DNA ligase (NEB), and residual linear DNA was degraded by Plasmid-Safe ATP-dependent DNase (Lucigen). In vitro cleavage reactions were performed with 250 ng of Plasmid-Safe-treated circularized DNA, 90 nM of Cas9 nuclease (NEB), Cas9 nuclease buffer (NEB) and 90 nM of synthetically modified sgRNA (Synthego) in a 100 μL volume. Cleaved products were A-tailed, ligated with a hairpin adapter (NEB), treated with USER enzyme (NEB) and amplified by PCR with barcoded universal primers NEBNext Multiplex Oligos for Illumina (NEB), using Kapa HiFi Polymerase (KAPA Biosystems). Libraries were sequenced with 150 bp paired-end reads on an Illumina MiSeq instrument. CIRCLE-seq data analyses were performed using open-source CIRCLE-seq analysis software (https://github.com/tsailabSJ/circleseq) using default parameters. The mouse genome mm10 (GRCm38) was used for alignment.

### Retro-orbital and intrafemoral injections
AAV was diluted into either 10 or 100 μL of sterile PBS for IF and intravenous RO injections, respectively. Thirty minutes before injections, GFP-on or GFP-on/Rag2$^{-/-}$ mice ($n = 1$–3, females, 15–24 weeks at harvest time) were injected subcutaneously with 5 mg/kg of Carprofen (OstiFen) (VetOne; MWI Veterinary Supply, Boise, ID) as an analgesic. Mice were anesthetized through the inhalation of 2% isoflurane (Fluriso) (VetOne; MWI Veterinary Supply, Boise, ID). To maximize systemic delivery, intravenous delivery was performed through combined RO and IF injections into three 12–14 week-old mice per delivery strategy. 100 μL AAV was injected into the retro-bulbar sinus using a 1 mL 29-gauge insulin syringe. For IF injection, a 27-gauge needle (World Precision Instruments, https://www.wpiinc.com/ Sarasota, FL) was initially used to gently puncture the patellar tendon. Subsequently, a 29-gauge syringe (Hamilton, Franklin, MA) containing 10 μL of AAV was inserted into the same puncture, and the solution was gently injected into the femur.

### In utero injections
In utero gene treatment (IUGT) was performed in pregnant GFP-on or Rag2$^{-/-}$ females mated with GFP-on males at embryonic day (E) 13.5–14.5. Thirty minutes before surgery, mice were injected subcutaneously with 0.5–1 mg/kg Buprenorphine ER (ZooPharm/ Wedgewood Pharmacy, Swedesboro, NJ) as an analgesic and pre-warmed fluids (0.9% NaCl) were given simultaneously with analgesics. Anesthesia was induced by inhalation of 3–4% isoflurane and maintained at 1–3% throughout the surgery. The surgical site was shaved, and the skin was disinfected using Betadine (Purdue Products, Stamford, CT). A midline laparotomy was then performed, and the uterine horns were externalized under sterile conditions. Using a 33-gauge syringe (World Precision Instruments, Sarasota, FL), 2–5 μL (2.7e10) of total vg of AAV were administered via intrahepatic injection into each fetus ($n = 8$–9, females/males). The uterus was returned to the abdomen, and incisions were closed in distinct anatomic layers with absorbable sutures. Mice were monitored daily and given Carprofen (5 mg/kg) 24 h post-surgery. Carprofen (5 mg/kg) or Buprenorphine ER (0.5–1 mg/kg) was then administered as needed based on pain assessment. Dams were kept on antibiotic water 0.1 mg/mL Enrofloxacin (Enrosite) (Veterinary Supply, Boise, ID) for 1 week post-surgery. After natural delivery at E20-E21 day of gestation, treated pups were imaged using a visualized using Xite-RB Fluorescent Imaging with 440–460 nm excitation and 500 nm longpass visualization (Nightsea, Lexington, MA) and kept with the mother until weaning. The pups were housed until scheduled post-mortem examination at 8, 16, or 20 weeks postnatally.

### Post-mortem examination of GFP-on mice treated with dual AAV9
A mixture of 100 mg/kg Ketamine (Vedco Inc., St Joseph, MO) and 10 mg/kg Xylaine (Akorn Animal Health, Lake Forest, IL) was administered intraperitoneally into treated and control GFP-on and EGFP$^+$ mice to induce deep anesthesia before the procedure. Mice were then perfused with ice-cold PBS, the right atrium was incised, and a 26-gauge syringe was inserted into the left ventricle to remove blood from organs to avoid autofluorescence. Whole-body imaging was performed by using a fluorescence flashlight system. Organs (heart, kidney, brain, skin, eye, liver, muscle, spleen, and lungs) were isolated to assess the GFP restoration by microscopy imaging, HTS, and immunofluorescence. EGFP expression was assessed using flow cytometry in the liver, spleen, bone marrow, and peripheral blood.

### Flow cytometry analysis
Bone marrow cells were flushed from the femurs of euthanized mice. The spleen was homogenized through a 70 μm cell strainer (Corning, Glendale, Arizona) into a petri dish (Corning, Glendale, Arizona). The remaining red blood cells (RBCs) were removed from the bone marrow cells, spleen cells, and peripheral blood cell suspensions with RBC lysis buffer (eBioscience, San Diego, CA). The liver was dissociated into single-cell suspensions by combining mechanical disruption with enzymatic digestion using a liver dissociation kit (Miltenyi Biotec, Auburn, AL) and the gentleMACS Dissociator according to the manufacturer's instructions. One million cells from peripheral blood, bone marrow, spleen, and liver were stained either with blue viability dye alone or with a panel of antibodies (Supplementary Table 2) for 20 min at room temperature. The samples were assessed using BD FACSymphony A5SE, and data were analyzed using FlowJo v10.

### Immunofluorescence
Tissues were fixed in 4% paraformaldehyde (Thermo Fisher, Houston, TX) overnight, immersed in 10 or 15%, and 30% sucrose for 24 h each, and embedded in Optimal Cutting Temperature (OCT) compound. Subsequently, tissues were sectioned at a thickness of 10 μm using a cryostat (Leica, Teaneck, NJ). Tissue sections were mounted onto 1% gelatin-coated slides (Fisher Scientific, Houston, TX), imaged for EGFP expression, and then stored at −80 °C until staining was performed.

Before staining, tissue sections were washed with PBS, permeabilized with a 0.01% Triton solution (Sigma-Aldrich, St. Louis, MO) for 20 min, and blocked with 5% bovine serum albumin (Sigma-Aldrich, St. Louis, MO) and 0.03% Sodium Azide for 1 h. Subsequently, tissue

sections were incubated with 1:1000 NeuN (MAB377A5, Sigma-Aldrich, St. Louis, MO), 1:100 HNF4a (MA1199, Invitrogen, Carlsbad, CA), or 1:200 Laminin (MA106821, Invitrogen, Carlsbad, CA) antibodies, overnight at 4 °C. Sections were washed with PBS and incubated with secondary antibodies at 4 °C for 1 h: 1:100 goat anti-mouse IgG (A21245, Invitrogen, Carlsbad, CA) or 1:200 goat anti-rat IgG AF647 (ab150167, Abcam, Cambridge, MA). Tissue sections were mounted using Fluoromount-G with DAPI (Invitrogen, Carlsbad, CA) and coverslipped. All images were captured at ×10 magnification using an Echo Revolve RVL2-K microscope (San Diego, CA).

### Whole-body fluorescence cryo-imaging

The hyperspectral fluorescence cryo-imaging system has been previously described[37]. Briefly, this instrument images frozen specimens during autonomous sectioning to produce high-resolution 3-D RGB and fluorescence image stacks of entire animals. In this study, whole animal specimens were prepared for imaging by submerging in OCT compound and freezing at −20 °C. Specimens were then mounted in the instrument and sectioned in 100 µm increments and imaged using brightfield and EGFP fluorescence channels. Specifically, brightfield images were acquired by illuminating the specimen with a white light LED and acquiring hyperspectral images between 400 and 720 nm in steps of 10 nm using a liquid crystal tunable filter (LCTF, CRI Varispec) The EGFP channel used a 470 nm excitation LED (Mightex, Toronto, ON, Canada) and acquired fluorescence images in 10 nm bands between 510 and 600 nm using the LCTF. The RGB and fluorescence image stacks were processed and rendered into volumes using 3DSlicer (http://www.slicer.org). 3DSlicer was also used to segment brain, liver, heart, skeletal muscle and spleen tissues and compute mean fluorescence values in each tissue.

### Statistics and reproducibility

Data are presented as mean and standard deviation (SD). The sample size and the statistical tests used for each experiment are described in the figure legends. No statistical methods were used to predetermine sample sizes. No data were excluded from the analyses. All HTS data were analyzed using an automated CRISPResso2 program that does not allow experimenter intervention, therefor experimenter was not blinded through data analysis. Animals were generally assigned a number that was subsequently unblinded after analysis. This was done for all the assays that were performed, except for HTS. For all in vitro experiments, conditions were assigned randomly across wells, and plate positioning is not expected to affect experimental outcomes. Allocation of animals was random and was performed by independent technicians. They assigned mice to different groups in a random fashion. Sex was considered in the study design, and both male and female mice were used for the studies, with approximately equal portions of each, which was selected by random distribution depending on the availability of animals. Disaggregated data for sex is included where available. Given the small cohort size with similar results between male and female mice, sex-based analyses were not reported. Statistical analyses were performed using GraphPad Prism 10 software.

### Reporting summary

Further information on research design is available in the Nature Portfolio Reporting Summary linked to this article.

## Data availability

The following plasmids used in this study have been deposited to Addgene (https://www.addgene.org) under Addgene IDs 239014, 239016, 239017. High-throughput DNA sequencing FASTQ files generated in this study have been deposited in the National Center of Biotechnology's Information Sequence Read Archive under BioProject "PRJNA1169082" [https://www.ncbi.nlm.nih.gov/bioproject/PRJNA1169082/]. Source data are provided with this paper.

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

## Acknowledgements

The authors thank Cynthia Klein, Mark Krampf, Stanford University, and Rendall R. Strawbridge, Dartmouth College, for their help with laboratory management. They thank Katie Ho, Ethan Haslett, and Aurora Connor for their invaluable assistance and technical expertise throughout this project. This study was funded by the Broad Institute Shark Tank Award to G.A.N., the NIH (R00 HL 163805 to G.A.N.; and UG3AI150551, U01AI142756, R35GM118062, RM1HG009490 to D.R.L.), and HHMI to D.R.L. This project was additionally funded by a gift from The Taube Stem Cell Pediatric Cancer Research Fund to A.D.C. This project was also supported by a grant from the Fanconi Cancer Foundation (formerly Fanconi Anemia Research Fund) to A.D.C.

## Author contributions

G.A.N., C.D., J.A.Q., M.D., T.C.M., D.R.L., and A.D.C. designed the study; G.A.N., C.D., J.A.Q., L.S., H.W., M.D., C.T.C., and B.P.Y. conducted experiments and analyzed data; F.M.Z., B.B., and T.L. performed immunofluorescence and analyzed data; C.Y.K., A.S., and S.C.D. performed whole-body cryo-macrotome imaging and analyzed data; J.A.Q., N.A., J.R.D., R.H., J.X., and G.G. produced ABE delivery reagents; C.D., J.A.Q., G.A.N., F.MZ., S.C.D., D.R.L., and A.D.C. wrote the manuscript with the input of all authors; G.A.N., D.R.L., and A.D.C. supervised this study.

## Competing interests

A.C. discloses financial interests in the following entities working in the rare genetic disease space: Beam Therapeutics, Editas Medicines, Fulcrum Therapeutics, Global Blood Therapeutics, Inograft Biotherapeutics, Land Medicine, Prime Medicine, and Spotlight Therapeutics. D.R.L. is a consultant and/or equity holder of Prime Medicine, Beam Therapeutics, Pairwise Plants, and nChroma Bio, companies that use gene editing or genome engineering. J.R.D. is a current employee and equity holder of Prime Medicine. D.R.L. is an inventor of base editors (US 20170121693). D.R.L. and J.R.D. are inventors of AAV vectors encoding base editors (US 20250064981). A.C. and S.C.D. have a financial interest in GV, a venture capital firm funded by Alphabet, and its portfolio. A.C. has received research funding from Rocket Pharmaceuticals, Jasper Therapeutics and STRM.Bio. S.C.D. has received research funding from Aera Therapeutics. None of these entities funded or were involved in this study. The remaining authors report no competing interests.
