## [Transparent Peer Review file · Nature Communications]

GFP-on mouse model for interrogation of in vivo gene editing

Corresponding Author: Professor Agnieszka Czechowicz

Version 0:

Reviewer comments:

Reviewer #1

(Remarks to the Author)

Here the authors have designed and evaluated a GFP-on mouse model that makes it easier to study in vivo gene editing. The data are interesting and supported by the evidence presented by the authors. The specific comments below should be considered before publication.

1 - Please expand how / why you chose the three potential candidates (Q70X, Q81X, Q95X). This will help scientists as they consider designing their own reporter systems

2 - Please comment on the sensitivity of the system (e.g., Figures 2e/f).

2b - In the discussion, please add more writing re: the sensitivity of the system relative to a therapeutic edit. For example, if a delivery system group saw X% editing using the system and then tried therapeutic editing for a disease-causing gene, would you expect them to see X% editing? More / less? Why?

3 - Apologies if I missed it, but are the authors planning to make the mouse available? They have every right now to, but if they are going to, it would be good to share how to access the mouse.

4 - In the discussion it would be helpful to understand if the authors expect any leakiness from the system. Many labs have a difficult time anticipating leakiness from Cre-based systems, so it would be good to understand how Cre-based leakiness differs from this system.

Reviewer #3

(Remarks to the Author)

Comments to Authors:

In this very interesting and important paper, Drs. Dib and colleagues describe the creation of a valuable new line of mice known as "GFP-on" mice, that harbor a single nucleotide mutation in 6 copies of an eGFP transgene. As such, these mice can be used to test and screen existing and newly developed genome editing (specifically base editors), with high sensitivity, specificity, and in relatively high throughput. Importantly, this new model also enables the assessment of in vivo targeting of the base editor delivery, at both the tissue and the single cell level, making it highly valuable for screening various viral vectors and nanoparticle-based delivery systems to determine which is optimal for achieving efficient editing in the cell type of interest. The authors describe very well designed experiments in vitro, in vivo in adult animals, and in vivo in fetal animals, proving the broad applicability of this new model for assessing base editing efficiency and targeting to various tissues. Importantly, the authors demonstrate successful base editing by both restoration of GFP expression and by sequencing. There is no doubt this new model fills a critical gap in our ability to accurately assess gene editing tools and predict their ability to mediate a therapeutic effect by achieving sufficient levels of editing in the necessary cell types while minimizing editing in off-target tissues and cell types. These studies have broad applicability and the impact this new model will exert on the field is likely to be substantial. There are, however, a few questions/issues the authors should address:

1. The authors state that they determined the copy number of GFP transgene using GAPDH as a reference gene. The mouse genome contains 285 GAPDH pseudogenes – could the authors comment on how they designed their ddPCR primers and probe to ensure only the true GAPDH gene was amplified, given that the sequence of many of the pseudogenes is almost identical to that of the true gene?

2. One would expect the editing efficiency to be fairly high on fibroblasts in vitro, when one can optimize all parameters, yet

the authors demonstrate an editing rate of only 2.6%. Moreover, they then go on to show ten-fold higher editing rates in some tissues in vivo. Is this a function of AAV9, i.e., does AAV9 not transduce fibroblasts very well, or do the authors have another explanation?

3. Can the authors comment on the finding that only 3 of the 5 injected fetuses exhibited GFP fluorescence? Was this likely a technical issue with the injection? Along these same lines, only 2 of the injected fetuses exhibited GFP fluorescence in their liver, one at 18% and the other at only 2.5%. This is a very wide range of editing efficiency (0-18%), considering all 3 fetuses received the same dose of the same AAV9 vector carrying the base editing machinery, and it raises some concerns regarding the ability of these model to tease out subtle differences between different delivery vectors and/or base editing platforms if this much animal-to-animal variation exists.

4. In Figures 3 and 4, there does not appear to be a very good correlation between the rate of editing determined by sequencing (panel C) and the level of eGFP fluorescence observed by confocal microscopy (panel B). While one might expect to see higher rates of editing than eGFP fluorescence, it is hard to understand how such low rates of editing would yield such widespread and intense levels of eGFP fluorescence. Could the authors comment on this apparent disconnect between the two methods of assessing editing efficiency?

Reviewer #4

(Remarks to the Author)

Reviewer Comments and Suggestions

The authors developed the GFP-on reporter mouse model carrying the EGFP Q81X mutation, providing a platform to study the efficiency of SpCas9-based adenine base editors (ABEs) delivered to specific tissues or cells. The authors validated their platform by delivering ABE8e-sgRNA packaged in AAV9 serotype vectors to the GFP-on mice, demonstrating GFP fluorescence correction in AAV9-targeted tissues. The model is also compatible with orthologue screening due to the presence of PAM sequences (NGG, NNGG, NNGRRT) required by SpABE8e, SauriABE8e, and SaABE8e near the EGFP Q81X mutation. However, additional data and more detailed analyses are needed to improve the overall quality and reliability of the manuscript. Below are the specific questions and suggestions:

Q1. In Fig. 1, the authors demonstrated the generation of the GFP-on mouse model by microinjecting BE4max-SpCas9-NG and sgRNA (GAAGCAGCACGACTTCTTCA) into zygotes across two generations. The final mice showed no bystander editing in the three copies of the EGFP gene. However, the potential off-target effects of the sgRNA on other genes and the absence of unexpected C-to-T substitutions or indels were not addressed. By validating the absence of such off-target effects in the generated mouse model, the reliability and broader utility of the model can be significantly enhanced.

Q2. The authors claim that the EGFP mutation allows targeting by SpABE8e, SauriABE8e, and SaABE8e at positions inducing stop codons, making the model adaptable to all three orthologues. However, validation using SauriABE8e packaged in AAV is missing. Providing such data would strengthen the authors' claims.

Q3. In Supplementary Figure 4, SpABE8e and SaABE8e delivered via AAV9 serotype were shown to induce similar GFP expression patterns in major organs, visualized through MIP images. However, differential expression was observed in specific regions of the brain and heart. For example, GFP expression was stronger in the midbrain for SpABE8e and in the olfactory bulb for SaABE8e. Is this variability due to differences between the base editors themselves or the delivery systems (e.g., Dual-AAV vs. Single-AAV)? Including a discussion on this would improve readers' understanding.

Q4. The authors attempted gene editing during fetal development by delivering AAV9-SpABE8e-sgRNA1 in utero. Does the injected AAV9 also affect maternal tissues?

Q5. Following Q4, does AAV9-SpABE8e-sgRNA1 induce any unintended edits in genes other than EGFP in the fetus?

Q6. Expanding the validation of the GFP-on model to include various AAV subtypes or target tissues would broaden the scope of the research and demonstrate the versatility of the platform.

Q7. Including comparative data with existing model systems (e.g., Cre-LoxP or traditional reporter systems) would highlight the advantages of this novel GFP-on model.

Q8. While AAV9 was the primary delivery method used in this study, the manuscript claims the platform can test "various delivery methods." Including results from alternative delivery methods (e.g., lipid nanoparticles or electroporation) would showcase the model's broader applicability and generalizability.

Version 1:

Reviewer comments:

Reviewer #1

(Remarks to the Author)

The authors have addressed all my comments.

Reviewer #3

(Remarks to the Author)

The authors have done a very thorough job of satisfactorily addressing all of the questions I raised, as well as those raised by the other Reviewers. I have no further questions or concerns.

Reviewer #4

(Remarks to the Author)

The authors' development of the GFP/on reporter mouse model and its use in evaluating the in vivo editing efficiency of SpCas9-based ABEs is a valuable contribution. In particular, the demonstration of GFP restoration in major organs following AAV9 mediated delivery of SpABE8e and SaABE8e highlights the practical utility of this model. The potential to overcome limitations of the conventional Cre/LoxP system is also noteworthy. However, several aspects could be further refined to enhance the completeness of the manuscript.

1. The authors state that the F0 mice were backcrossed with WT C57BL/6 mice for two generations to minimize off-target effects. However, given the nature of the ABE system used in this study, unintended A-to-G edits at non-target sites may still be possible. To strengthen the reliability of the findings, the authors could consider performing amplicon sequencing or targeted deep sequencing for the top ranked off-target candidates identified in the CIRCLE seq analysis.

2. While the absence of in vivo validation for SauriABE8e is understandable given the study's scope, it would be beneficial to provide supporting evidence for its applicability to the GFP/on model. If in vivo validation is not feasible, an alternative approach could involve in vitro experiments using HEK293T cells or GFP/on mouse-derived cells to demonstrate editing efficiency. If additional experiments are impractical, discussing existing literature on efficiency and limitations of SauriABE8e would strengthen the manuscript's rationale.

3. In Supplementary Figure 4, SpABE8e and SaABE8e exhibited different editing patterns in specific tissues, particularly in the brain and heart. The authors attribute this to differences in single vs. dual AAV delivery. However, another potential explanation could be tissue specific AAV9 tropism and cellular targeting efficiency. Recent studies suggest that AAV9 exhibits differential efficiency in distinct brain regions (e.g., midbrain vs. olfactory bulb). Adding a discussion on this aspect would help clarify the observed variability and improve the interpretation of results.

4. The study reports in utero AAV9 delivery for fetal gene editing but does not analyze potential AAV9 dissemination to maternal tissues. The authors argue that low fetal editing efficiency suggests minimal maternal impact, but previous studies indicate that AAV9 can cross the placenta and enter maternal circulation. If direct maternal tissue analysis is not feasible, referencing existing studies on AAV9 biodistribution in pregnancy would provide a more comprehensive discussion.

5. The manuscript emphasizes that the GFP/on model is more sensitive to detecting lower-efficiency edits compared to Cre/LoxP systems. While this is a valuable point, experimental evidence demonstrating the limitations of Cre/LoxP systems in evaluating AAV-mediated delivery efficiency would enhance the argument. A direct comparison between Cre/LoxP-based GFP activation models and the GFP/on model (either through experimental data or a literature-based analysis) would further emphasize the novelty and significance of this study.

Version 2:

Reviewer comments:

Reviewer #4

(Remarks to the Author)

The authors have sufficiently addressed all my comments. I have no further questions or concerns.

POINT BY POINT RESPONSES:

=====

Reviewer #1 (Remarks to the Author):

Here the authors have designed and evaluated a GFP-on mouse model that makes it easier to study in vivo gene editing. The data are interesting and supported by the evidence presented by the authors. The specific comments below should be considered before publication.

We thank the reviewer for recognizing the importance of this work and for their additional comments which we have addressed below and in the revised manuscript.

1 - Please expand how / why you chose the three potential candidates (Q70X, Q81X, Q95X). This will help scientists as they consider designing their own reporter systems.

We have adjusted lines 126-129 to better address this question. It now reads:

“We chose three potential candidates near the 5’ end of the EGFP ORF ~~with and containing~~ a target nucleotide positioned 12-16 bases upstream of an NGG PAM site (Q70X, Q81X, Q95X) (Supplementary Fig 1).”

2 - Please comment on the sensitivity of the system (e.g., Figures 2e/f).

The sensitivity of the GFP-on model system is enabled by the ability to detect GFP in even single cells via various readily available technologies such as flow cytometry. The high degree of correlation between edited genotype and % GFP+ phenotype at the 3% editing level (Miseq limit of detection is about 0.1%) in vitro demonstrates that the data shown in Fig 2e/f is not below the limit of detection for EGFP fluorescence. The GFP-on model can accurately reflect relative changes in base editing efficiency at least as low and likely lower than 3%.

2b - In the discussion, please add more writing re: the sensitivity of the system relative to a therapeutic edit. For example, if a delivery system group saw X% editing using the system and then tried therapeutic editing for a disease-causing gene, would you expect them to see X% editing? More / less? Why?

Relative to a recessive therapeutic target (copy number 2) or dominant negative mutation (copy number 1), the copy high copy number in the GFP-on mouse will lead to lower editing efficiencies than most therapeutic edits due to the high count of total alleles. However, we would expect closer to a 1:1 relationship in the percent of edited cells, which would be the main readout for typical users of this model. Hence, this is a very sensitive model that can detect even low-level editing while also establishing the frequency of cells that can be edited across tissues. This has been added to the discussion [lines 333-337].

3 - Apologies if I missed it, but are the authors planning to make the mouse available? They have every right now to, but if they are going to, it would be good to share how to access the mouse.

Once published, we absolutely intend to share this mouse model upon request. We have included this wording within the manuscript to better clarify this [line 343].

4 - In the discussion it would be helpful to understand if the authors expect any leakiness from the system. Many labs have a difficult time anticipating leakiness from Cre-based systems, so it would be good to understand how Cre-based leakiness differs from this system.

In addition to the data shown in Fig. 1f and Supp. Fig. 4 which show very low leakiness using flow cytometry and 3D Cryo imaging, respectively, we have added a discussion on the mechanistic difference between Cre systems and the GFP-on mouse [lines 281-295].

Reviewer #3 (Remarks to the Author):

Comments to Authors:

In this very interesting and important paper, Drs. Dib and colleagues describe the creation of a valuable new line of mice known as “GFP-on” mice, that harbor a single nucleotide mutation in 6 copies of an eGFP transgene. As such, these mice can be used to test and screen existing and newly developed genome editing (specifically base editors), with high sensitivity, specificity, and in relatively high throughput. Importantly, this new model also enables the assessment of in vivo targeting of the base editor delivery, at both the tissue and the single cell level, making it highly valuable for screening various viral vectors and nanoparticle-based delivery systems to determine which is optimal for achieving efficient editing in the cell type of interest. The authors describe very well designed experiments in vitro, in vivo in adult animals, and in vivo in fetal animals, proving the broad applicability of this new model for assessing base editing efficiency and targeting to various tissues. Importantly, the authors demonstrate successful base editing by both restoration of GFP expression and by sequencing. There is no doubt this new model fills a critical gap in our ability to accurately assess gene editing tools and predict their ability to mediate a therapeutic effect by achieving sufficient levels of editing in the necessary cell types while minimizing editing in off-target tissues and cell types. These studies have broad applicability and the impact this new model will exert on the field is likely to be substantial. There are, however, a few questions/issues the authors should address:

We thank the reviewer for recognizing the value of our work and of this new line of GFP-on mice which allow for testing of new gene editing tools with high sensitivity, specificity and throughput. We believe these tools and mice will be of high utility to the scientific community. We appreciate the additional questions that we have addressed below and in the revised manuscript.

1. The authors state that they determined the copy number of GFP transgene using GAPDH as a reference gene. The mouse genome contains 285 GAPDH pseudogenes – could the authors comment on how they designed their ddPCR primers and probe to ensure only the true GAPDH gene was amplified, given that the sequence of many of the pseudogenes is almost identical to that of the true gene?

The digital PCR assay used to determine GAPDH position was purchased as a stock primer/probe mix from Bio-Rad (Mouse GAPDH ddPCR Copy Number Assay (10042958; Assay ID = dMmuCNS133125454)). They use a proprietary algorithm to design their primer/probe mixes and were not willing to share any details of this design upon our request. However, while they do not share the exact sequences of the primers and probe, they do specify the sequence context that is amplified by the assay, which is:
CCAATAAAGATACATGCACAAAAGTTGATTGAGCCTGCTTCACCTCCCCATACACACC
CTCCCTCCCCAACACCGCATTAAAACCAAGGAGAGGTGGGTGCAGCGAACTTTATT
GATGGTAT.

We entered this sequence into BLAST to search the mouse genome for homology. The results of that BLAST search identified only the target site as having 100% homology to this sequence. One other region in the mouse genome had similarity to 75% of the query sequence, and all other regions in the mouse genome matched to 26% or less of the query sequence. The one region with 75% homology did not detect homology in the first 30 nucleotides of the targeted sequence, which should prevent binding of the forward primer and prevent a positive dPCR signal. To clarify this point in the text, we modified the methods to include the Bio-Rad assay ID and described our BLAST search that confirmed there is only a single target of this assay in the mouse reference genome [lines 382-390].

2. One would expect the editing efficiency to be fairly high on fibroblasts in vitro, when one can optimize all parameters, yet the authors demonstrate an editing rate of only 2.6%. Moreover, they then go on to show ten-fold higher editing rates in some tissues in vivo. Is this a function of AAV9, i.e., does AAV9 not transduce fibroblasts very well, or do the authors have another explanation?

AAV9 is known to be far less efficient as an in vitro delivery vehicle, especially in fibroblasts [1], though we considered it both critical to show validation of the GFP-on mechanism ex vivo using AAV9 and an important demonstration for users of the mouse model to be able to edit primary mouse cells with an accurate benchmark of the expected editing efficiency. The AAV serotypes typically used for in vitro transduction to which the reviewer may be referring are AAV2 [2] and AAV6 [3,4].

3. Can the authors comment on the finding that only 3 of the 5 injected fetuses exhibited GFP fluorescence? Was this likely a technical issue with the injection? Along these same lines, only 2 of the injected fetuses exhibited GFP fluorescence in their liver, one at 18% and the other at only 2.5%. This is a very wide range of editing efficiency (0-18%), considering all 3 fetuses received the same dose of the same AAV9 vector carrying the base editing machinery, and it raises some concerns regarding the ability of these model to tease out subtle differences between different delivery vectors and/or base editing platforms if this much animal-to-animal variation exists.

We believe the variability in editing amongst the different treated fetuses is due to the technical challenges associated with the injection as the volume administered was quite small in these very small fetal mice. In a clinical setting for in utero treatment, this would be performed under ultrasound guidance with a larger volume. Moreover, the in utero treatment experiments were primarily included in this manuscript to showcase the utility of this model for different types of delivery methods, and while these experiments demonstrate the proof-of-concept for in utero treatment they are not intended to validate the use of dual AAV9 delivery in the utero setting. Additionally, the variability may be due to the use of dual AAV9 vectors which require the co-delivery of both vectors and future in utero experiments intend to explore the use of single AAV9 as well as other delivery modalities in larger cohorts of animals.

4. In Figures 3 and 4, there does not appear to be a very good correlation between the rate of editing determined by sequencing (panel C) and the level of eGFP fluorescence observed by confocal microscopy (panel B). While one might expect to see higher rates of editing than eGFP fluorescence, it is hard to understand how such low rates of editing would yield such widespread and intense levels of eGFP fluorescence. Could the authors comment on this apparent disconnect between the two methods of assessing editing efficiency?

We interpret the divergence between % GFP+ cells and editing efficiency to be due to the high copy number of the integrated GFP gene. With a copy number of six in homozygous mice and assuming only one gene copy needs to be corrected to observe GFP fluorescence, we would expect

editing efficiency to be an underestimate rather than an overestimate of the % GFP+ cells as Reviewer 3 reasons.

As a demonstration, a typical recessive gene variant with copy number of two, the minimum editing efficiency required to achieve corrected protein is 50% (to restore heterozygosity). However, in this GFP^{pm/pm} mouse, the high copy number suppresses this value such that editing one of six available alleles to restore GFP expression in the cell would require at a minimum editing efficiency of just 16.67%.

This important point has been added to the discussion section [lines 333-337] in which we note that “The elevated copy number...decreases the percent of edited alleles in this model relative to the percent of edited cells. This effect becomes most apparent in vivo due to the challenge of achieving high transgene expression in vivo relative to ex vivo cultured cells. Therefore, the percent of GFP positive cells is a more translatable indicator of base editing efficiencies across other protospacers.”

Reviewer #4 (Remarks to the Author):

Reviewer Comments and Suggestions

The authors developed the GFP-on reporter mouse model carrying the EGFP Q81X mutation, providing a platform to study the efficiency of SpCas9-based adenine base editors (ABEs) delivered to specific tissues or cells. The authors validated their platform by delivering ABE8e-sgRNA packaged in AAV9 serotype vectors to the GFP-on mice, demonstrating GFP fluorescence correction in AAV9-targeted tissues. The model is also compatible with orthologue screening due to the presence of PAM sequences (NGG, NNGG, NNGRRT) required by SpABE8e, SauriABE8e, and SaABE8e near the EGFP Q81X mutation. However, additional data and more detailed analyses are needed to improve the overall quality and reliability of the manuscript. Below are the specific questions and suggestions:

We appreciate the reviewer’s detailed review of our work with additional questions and suggestions that we have addressed below and have led to a stronger, revised manuscript.

Q1. In Fig. 1, the authors demonstrated the generation of the GFP-on mouse model by microinjecting BE4max-SpCas9-NG and sgRNA (GAAGCAGCACGACTTCTTCA) into zygotes across two generations. The final mice showed no bystander editing in the three copies of the EGFP gene. However, the potential off-target effects of the sgRNA on other genes and the absence of unexpected C-to-T substitutions or indels were not addressed. By validating the absence of such off-target effects in the generated mouse model, the reliability and broader utility of the model can be significantly enhanced.

Following standard practice in CRISPR-Cas9 modified mouse lines, the F0 generation was backcrossed with WT C57BL/6 mice for two generations before the start of this study. This process selects for the presence of the EGFP gene in subsequent generations and breeds out any potential off-target edits. It is for this reason that whole genome sequencing is not typically performed on every CRISPR-Cas9 edited mouse model.

Additionally, we did not observe any notably phenotypic changes between the GFP-on mouse and the parent EGFP+ mouse, nor observed any difference by flow cytometric analysis of bone marrow or peripheral blood.

Q2. The authors claim that the EGFP mutation allows targeting by SpABE8e, SauriABE8e, and SaABE8e at positions inducing stop codons, making the model adaptable to all three orthologues. However, validation using SauriABE8e packaged in AAV is missing. Providing such data would strengthen the authors' claims.

To our knowledge, there are no literature examples of single SauriABE-AAV9 in vivo editing. The only data comes in the context of AAV8 [5]. As this paper is intended to show utility of this novel mouse model and not develop new delivery vehicles, we see developing SauriABE-AAV9 to be beyond the scope of this study. However, for this ABE site, SauriABE8e and SaABE8e would use identical sgRNAs (Supp. Fig. 1), so the validation of editing in the SaABE8e context lends credence to the use of the identical sgRNA to be used with SauriABE8e, albeit with a shorter PAM recognition sequence in the case of Sauri.

We considered it important to explicitly mention SauriABE so future users are aware of 1) the utility of the model in studies concerning SauriABE delivery and 2) to further rationalize our choice to use the Q81X mutation which may help inform others seeking to make similar models.

Q3. In Supplementary Figure 4, SpABE8e and SaABE8e delivered via AAV9 serotype were shown to induce similar GFP expression patterns in major organs, visualized through MIP images. However, differential expression was observed in specific regions of the brain and heart. For example, GFP expression was stronger in the midbrain for SpABE8e and in the olfactory bulb for SaABE8e. Is this variability due to differences between the base editors themselves or the delivery systems (e.g., Dual-AAV vs. Single-AAV)? Including a discussion on this would improve readers' understanding.

Single ABE-AAV vectors would be expected to show a more uniform editing profile across all tissues since each transduction event packages the entire editing system, thus removing the requirement for multiple transduction events. In our data (Supp. Figs. 4, 5), the single SaABE-AAV9 vector shows less relative editing in the major organs (liver, spleen, and skeletal muscle) than the dual SpABE-AAV9 vector at equivalent dose likely due to the overall lower editing efficiency of SaABE8e than SpABE8e. The single AAV shows dramatically higher editing efficiency in the heart and to a lesser extent the brain. Both tissues are less accessible from a biodistribution perspective following retro-orbital (RO) injection and may benefit from a single vector. Previous studies by our lab have attributed this phenomenon to single AAV rather than cell type specific activities of the Cas9 ortholog by delivering SaABE-AAV9 in both single and dual formats and showing increased editing in the heart and muscle for single AAV relative to the dual vector at an equivalent dose [5]. We would also note that while direct comparison of SpABE8e and SaABE8e is complicated by the fact that SpABE8e cannot be packaged in a single AAV and conversely SaABE8e suffers dramatic losses from intein splitting.

Similarly, the midbrain and cerebellum are more anatomically accessible than the olfactory bulb. While more replicates are necessary to support such a detailed observation, we do observe enriched editing in the olfactory bulb by single AAV9 delivery. Ultimately, such detailed observations require further exploration which we intend to report on in follow-up investigations, and this also motivate us to share this mouse model now with other labs that may have further interest and ability to perform such work.

Q4. The authors attempted gene editing during fetal development by delivering AAV9-SpABE8e-sgRNA1 in utero. Does the injected AAV9 also affect maternal tissues?

While we intend to perform future follow-on experiments to further detail findings related to exploration of different routes of administration, doses, and delivery vehicles, the intention of this manuscript is to explain, validate, and publicize the GFP-on mouse model to make these studies easier for many groups and to ease translation of in vivo base editing studies. As a proof-of-concept study, we did not analyze or save maternal tissues for this analysis.

However, we would expect there to be low to no-expression in maternal tissues, given that even in some of the fetuses that had direct administration of the AAV9, no EGFP expression was observed. The high variability in fetal editing is likely caused by the difficult injection technique into small fetal mice. Additionally, we have selected a viral dose tolerable for the fetal mouse which would be a much lower per mass dose for the mother were any AAV to escape the uterus and thus do not expect editing of maternal tissues.

Q5. Following Q4, does AAV9-SpABE8e-sgRNA1 induce any unintended edits in genes other than EGFP in the fetus?

While off-target editing is certainly a reality of any base editing approach, the use case for this mouse model is mainly for the EGFP phenotype as a reporter of delivery efficiency. In such biodistribution experiments, we would expect the phenotypic effects of off-target editing to be minimal given the short timeline of the experiments. Additionally, relative to Cre models, off target activity is likely much less perturbative in the base-editable GFP-on mouse whereas the former has been shown to cause off-target recombination of pseudo-LoxP genes in mice, one of which at 98.7% the efficiency of the WT LoxP sequence [6].

We do recognize the importance of considering the Cas9-dependent off-target transition mutations that may occur and have now provided a CIRCLE-seq nomination of candidate off-target loci as a new supplemental figure to the manuscript (Supp. Fig. 3). Please note, CIRCLE-seq was performed in WT mouse cells, thus the on-target EGFP locus does not appear on the allele chart. Further, no significant off-target editing was detected at the top 29 off-target loci nominated by CIRCLE-seq in genomic DNA extracted from cells treated with AAV9 either ex vivo or in vivo as which is showcased in the new supplemental figure (Supp Fig 4) [lines 187-190].

Q6. Expanding the validation of the GFP-on model to include various AAV subtypes or target tissues would broaden the scope of the research and demonstrate the versatility of the platform.

We aspire to perform and report observations such as the biodistribution of all commonly used AAV serotypes in follow on work, though this is outside the scope of the current study. This work is intended to demonstrate the utility of the mouse model in evaluating base editing activity across tissues rather than compare delivery efficiency of various vectors. Once published, we intend to share this mouse models with others in the scientific community that are interested in exploring various diverse delivery modalities and routes of administration. We have now specifically commented on the availability of this new mouse model [line 343].

Q7. Including comparative data with existing model systems (e.g., Cre-LoxP or traditional reporter systems) would highlight the advantages of this novel GFP-on model.

We have added a discussion paragraph to detail the expected improvements from the existing Cre-LoxP model systems to the GFP-on mouse [lines 281-295]. A key point we intend to make is that the field has routinely been misled by hyper-efficient Cre recombinase reporter systems which do not allow sufficient resolution of low-efficiency and high-efficiency delivery vehicles. This unfortunate pattern demands a reporter model compatible with therapeutically relevant cargoes.

We consider both the extensive literature on the Cre-LoxP fluorescent reporter mice and the demonstrated challenge of comparing Cre and base editor delivery that has necessitated the creation of the GFP-on mouse as sufficient rationale for the data presented on the GFP-on mouse to stand alone in this manuscript without direct comparison to a second model.

Q8. While AAV9 was the primary delivery method used in this study, the manuscript claims the platform can test "various delivery methods." Including results from alternative delivery methods (e.g., lipid nanoparticles or electroporation) would showcase the model's broader applicability and generalizability.

RNA electroporation was used to validate the sgRNAs prior to in vivo delivery (Fig. 2b, 2c) addressing this reviewer's request. In the future, we hope that this mouse model can enable us and others to make direct comparisons between various new and existing delivery vectors, though the scope of this manuscript is to report proof-of-concept of this mouse model. Additional studies would delay publication of this manuscript and slow dissemination of this mouse model which we believe is invaluable to the field. That said, we have performed preliminary ex vivo experiments showcasing that ABE8e-loaded engineered virus-like particles (eVLPs) [7] can efficiently base edit immortalized fibroblasts and Kit⁺ bone marrow cells from the GFP-on mouse. Further in vivo data comparing the biodistribution of various delivery modalities will be evaluated in follow-up studies leveraging the GFP-on mouse.

References:

- [1] Ellis, B.L., Hirsch, M.L., Barker, J.C. *et al.* A survey of *ex vivo/in vitro* transduction efficiency of mammalian primary cells and cell lines with Nine natural adeno-associated virus (AAV1-9) and one engineered adeno-associated virus serotype. *Virology* 10, 74 (2013). <https://doi.org/10.1186/1743-422X-10-74>
- [2] Krooss, S. A. *et al.* Ex vivo/in vivo gene editing in hepatocytes using 'all-in-one' CRISPR-adeno-associated virus vectors with a self-linearizing repair template. *iScience* 23, 100764 (2020).
- [3] Wilkinson, A.C., Dever, D.P., Baik, R. *et al.* Cas9-AAV6 gene correction of beta-globin in autologous HSCs improves sickle cell disease erythropoiesis in mice. *Nat Commun* 12, 686 (2021).
- [4] Pavani, G., Laurent, M., Fabiano, A. *et al.* Ex vivo editing of human hematopoietic stem cells for erythroid expression of therapeutic proteins. *Nat Commun* 11, 3778 (2020).
- [5] Davis, J.R *et al.* Efficient *In Vivo* Delivery of a Suite of Adenine Base Editors with Single Adeno-Associated Viruses with Size-Optimized Genomes Encoding Compact Adenine Base Editors. *Nat. Biomed. Eng* 6, 1272-1283 (2022).

[6] Thyagarajan B, Guimaraes MJ, Groth AC, Calos MP. Mammalian genomes contain active recombinase recognition sites. *Gene*. 2000;244:47–54.

[7] Banskota, S. *et al.* Engineered virus-like particles for efficient in vivo delivery of therapeutic proteins. *Cell* 185, 250-265.e16 (2022).

POINT BY POINT RESPONSES:

=====

Reviewer #1 & 3 (Remarks to the Author):

The authors have addressed all my comments.

The authors have done a very thorough job of satisfactorily addressing all of the questions I raised, as well as those raised by the other Reviewers. I have no further questions or concerns.

We thank the reviewers for appreciating our substantial revisions that have strengthen the manuscript and addressed these reviewers' concerns.

Reviewer #4 (Remarks to the Author):

The authors' development of the GFP/on reporter mouse model and its use in evaluating the in vivo editing efficiency of SpCas9-based ABEs is a valuable contribution. In particular, the demonstration of GFP restoration in major organs following AAV9 mediated delivery of SpABE8e and SaABE8e highlights the practical utility of this model. The potential to overcome limitations of the conventional Cre/LoxP system is also noteworthy. However, several aspects could be further refined to enhance the completeness of the manuscript.

We thank the reviewer for recognizing the value of our work and of this new line of GFP-on mice which allow for testing of new gene editing tools with high sensitivity, specificity and throughput. We believe these tools and mice will be of high utility to the scientific community. We appreciate the additional questions that we have addressed below and in the revised manuscript.

1. The authors state that the F0 mice were backcrossed with WT C57BL/6 mice for two generations to minimize off-target effects. However, given the nature of the ABE system used in this study, unintended A-to-G edits at non-target sites may still be possible. To strengthen the reliability of the findings, the authors could consider performing amplicon sequencing or targeted deep sequencing for the top ranked off-target candidates identified in the CIRCLE seq analysis.

We agreed with the reviewer's previous feedback that off-target characterization of the ABE system would improve the interpretation of the edit. As such, we performed targeted amplicon sequencing on the top ranked off-target candidates for the ABE system identified by CIRCLE seq. The results can be found in Supplementary Fig. 4.

2. While the absence of in vivo validation for SauriABE8e is understandable given the study's scope, it would be beneficial to provide supporting evidence for its applicability to the GFP/on model. If in vivo validation is not feasible, an alternative approach could involve in vitro experiments using HEK293T cells or GFP/on mouse-derived cells to demonstrate editing efficiency. If additional experiments are impractical, discussing existing literature on efficiency and limitations of SauriABE8e would strengthen the manuscript's rationale.

Ultimately, we anticipate that users of this model will choose to study the ABE activity of an editor that they intend to use at another locus to impart a desired phenotype. In this case, the user's test locus will determine which editor to use, and the broad PAM compatibility of this model allows multiple editors to correct EGFP^{pm} mutation. To emphasize this logic, we have added the following sentence to the manuscript:

Lines 329-332: "...this model can be targeted by *S. aureus* and *S. auricularis* ABEs, which are small enough to be packaged into a single AAV—thereby broadening the delivery methods suitable for the GFP-on model to match the current suite of therapeutic base editors."

3. In Supplementary Figure 4, SpABE8e and SaABE8e exhibited different editing patterns in specific tissues, particularly in the brain and heart. The authors attribute this to differences in single vs. dual AAV delivery. However, another potential explanation could be tissue specific AAV9 tropism and cellular targeting efficiency. Recent studies suggest that AAV9 exhibits differential efficiency in distinct brain regions (e.g., midbrain vs. olfactory bulb). Adding a discussion on this aspect would help clarify the observed variability and improve the interpretation of results.

Since all editors in this study were delivered using AAV9, we controlled for changes in the tropism of the delivery vehicle. The remaining variability arises from one of several factors which the GFP-on mouse is uniquely suited to resolve. For example, the single AAV systems being more capable of editing poorly transduced tissues as they do not need multiple transduction events to reconstitute an active ABE. Alternatively, there may be cellular cofactors that promote the activity of certain Cas variants over others and this effect may be cell type-specific. We are motivated to report these effects in future studies.

4. The study reports in utero AAV9 delivery for fetal gene editing but does not analyze potential AAV9 dissemination to maternal tissues. The authors argue that low fetal editing efficiency suggests minimal maternal impact, but previous studies indicate that AAV9 can cross the placenta and enter maternal circulation. If direct maternal tissue analysis is not feasible, referencing existing studies on AAV9 biodistribution in pregnancy would provide a more comprehensive discussion.

To make the readers aware of studies regarding the expression of AAV transgenes in maternal tissues following in utero injections, we have added the following sentence to discussion:

Lines 308-312: "While not assessed in this study, prior in utero AAV9 treatments have reported minimal leakage into maternal tissues with multiple studies in pregnant mice and lambs finding no detectable transgene expression in maternal tissues, though elevated bilirubin and anti-AAV antibody levels have occasionally been observed indicating some low level of exposure is possible.¹⁻⁴"

However, in the absence of active transgene expression, maternal tissues would remain GFP-negative in this model. Furthermore, these studies typically used AAV delivery of a fluorescent transgene, so less maternal editing would be observed than maternal transgene expression, further supporting our claim for minimal maternal editing efficiency.

5. The manuscript emphasizes that the GFP/on model is more sensitive to detecting lower-efficiency edits compared to Cre/LoxP systems. While this is a valuable point, experimental evidence demonstrating the limitations of Cre/LoxP systems in evaluating AAV-mediated delivery efficiency would enhance the argument. A direct comparison between Cre/LoxP-based GFP activation models and the GFP/on model

(either through experimental data or a literature-based analysis) would further emphasize the novelty and significance of this study.

In lines 290-297, we discuss how the Cre/LoxP system is actually more sensitive to detecting low-efficiency transduction events, but we reason that this leads to issues of translatability. The Cre/LoxP system is actually too sensitive meaning that delivery vehicles tested may trigger efficient recombination, but do not offer robust enough transduction to support genome editing. This has been observed by numerous studies.⁵⁻⁷

References:

1. Borges, B. *et al.* Prenatal AAV9-GFP administration in fetal lambs results in transduction of female germ cells and maternal exposure to virus. *Mol. Ther. - Methods Clin. Dev.* **32**, 101263 (2024).
2. Mattar, C. N. Z., Chew, W. L. & Lai, P. S. Embryo and fetal gene editing: Technical challenges and progress toward clinical applications. *Mol. Ther. - Methods Clin. Dev.* **32**, 101229 (2024).
3. Bose, S. K. *et al.* In utero adenine base editing corrects multi-organ pathology in a lethal lysosomal storage disease. *Nat. Commun.* **12**, 4291 (2021).
4. Rossidis, A. C. *et al.* In utero CRISPR-mediated therapeutic editing of metabolic genes. *Nat. Med.* **24**, 1513–1518 (2018).
5. Breda, L. *et al.* In vivo hematopoietic stem cell modification by mRNA delivery. *Science* **381**, 436–443 (2023).
6. Wei, T., Cheng, Q., Min, Y.-L., Olson, E. N. & Siegwart, D. J. Systemic nanoparticle delivery of CRISPR-Cas9 ribonucleoproteins for effective tissue specific genome editing. *Nat. Commun.* **11**, 3232 (2020).
7. Li, B. *et al.* Combinatorial design of nanoparticles for pulmonary mRNA delivery and genome editing. *Nat. Biotechnol.* **41**, 1410–1415 (2023).